# Structural basis of *Fusobacterium nucleatum* adhesin Fap2 interaction with receptors on cancer and immune cells

Felix Schöpf [1,2], Gian L. Marongiu[1,2], Klaudia Milaj[1,6], Thiemo Sprink[3,4], Judith Kikhney[5], Annette Moter[5] & Daniel Roderer [1] ✉

*Fusobacterium nucleatum* is overrepresented in the colon microbiome of colorectal cancer patients and has been associated with tumor growth enhancement and metastasis. A pivotal pathogenic factor, the autotransporter adhesin Fap2, facilitates association to cancer and immune cells via the receptors Gal-GalNAc and TIGIT, respectively, leading to deactivation of immune cells. Mechanistic details of the Fap2/TIGIT interaction remain elusive as no structural data are available. Here, we report a system to recombinantly express functional Fap2 on the *Escherichia coli* surface, which interacts with Gal-GalNAc on cancer cells and with purified TIGIT with submicromolar affinity. Cryo-EM structures of Fap2, alone and in complex with TIGIT, show that the elongated ~50 nm long Fap2 extracellular region binds to TIGIT on its membrane-distal tip via an extension of a β-helix domain. Moreover, by combining structure predictions, cryo-EM, docking and molecular dynamics simulations, we identified a binding pit for Gal-GalNAc on the tip of Fap2.

The human intestinal microbiome (IM) comprises up to $10^{14}$ microbes, a higher number than human cells in the body[1]. With estimated 22 million genes[2], the genetic diversity of the IM is orders of magnitude higher than the human genome. The composition of the IM contributes largely to the host's metabolism and health by providing different secreted bacterial enzymes, metabolites and lipids[3,4]. Pathological modulations of the IM, termed dysbiosis, have a severe effect on the intestinal system's health. Such dysbiosis is one of the key determinants for the prevalence of colorectal cancer (CRC), the third-most common cancer worldwide and the second-most cause of cancer related death[5]. 1.9 million new CRC cases and 935,000 deaths were estimated globally for 2020[6], with incidences consistently rising worldwide[7,8]. The over-representation of particular microbes in the IM can cause the onset and drive progression of CRC through various mechanisms, such as causing DNA damage through toxins[9], growth stimulation of tumor cells through the activation of signal transduction pathways, or down-regulation of the immune system[10]. *Fusobacterium nucleatum* (Fn), a rod-shaped, anaerobic Gram-negative, has been demonstrated to act via the latter two mechanisms[11–13]. Fn has been found over-represented in the IMs of CRC patients in numerous studies[14–20], and Fn has therefore been suggested as a biomarker for CRC[21]. Moreover, Fn has been shown to drive breast cancer growth[22], and to be associated with preterm birth, stillbirth[23], and periodontal disease[24], highlighting the clinical relevance of this understudied pathogen.

In contrast to other important CRC-driving microbiota, e.g., genotoxic *Escherichia coli* and *Bacteroides fragilis*[25], Fn encodes no known secreted toxins[26]. Instead, Fn produces numerous outer membrane (OM) anchored adhesins that are utilized to attach to other microbes in biofilms[27,28] or directly to host cells[29–31]. The latter causes tumor growth stimulation, e.g., through the interaction of the adhesin FadA to E-cadherin and subsequent activation of β-catenin translocation to the

[1]Leibniz-Forschungsinstitut für Molekulare Pharmakologie (FMP), Berlin, Germany. [2]Division Biology, Chemistry, Pharmacy, Freie Universität Berlin, Berlin, Germany. [3]Core Facility for Cryo Electron Microscopy of the Charité—Universitätsmedizin Berlin at the Max Delbrück Center, Berlin, Germany. [4]Max Delbrück Center for Molecular Medicine, Technology Platform Cryo-EM, Berlin, Germany. [5]Institute of Medical Microbiology and Virology, University of Leipzig Medical Center, Leipzig, Germany. [6]Present address: Faculty of Science, University of Potsdam, Potsdam, OT Golm, Germany. ✉e-mail: roderer@fmp-berlin.de

nucleus[11,32], or by interaction with the T-cell immunoreceptor with immunoglobulin and ITIM domains (TIGIT) on T cells or natural killer (NK) cells via the adhesin Fap2[33].

Fap2 is a type Va autotransporter OM protein[34] and with about 400 kDa the largest single protein in Fn[35]. It is a bifunctional adhesin through which Fn binds to tumors via Gal-GalNAc[36], a glycan that is abundant on and specific for cancer cells in the colon[37], and to immune cells, specifically NK cells, via TIGIT[33]. The combination of both interactions makes Fap2 a key molecule in Fn-driven CRC, as it mediates tumor colonization independently of the level of E-cadherin expression on tumor cells[36] as well as clearance of the tumor from active NK cells[38]. The molecular mechanisms underlying the binding of Fap2 to the two receptors remain however unknown because a high-resolution structure of Fap2 has not yet been identified.

Here we report an expression strategy for the recombinant production of Fap2 from *F. nucleatum subsp. nucleatum* ATCC23726 on the OM surface of *E. coli*, resulting in the production of functional Fap2 extracellular domain (ECD) that binds to both receptors, Gal-GalNAc and TIGIT. Structural analysis of the Fap2-ECD by cryogenic electron microscopy (cryo-EM) in combination with AlphaFold2 structure prediction reveals a 47 nm long rod-shaped β-helix with a matchstick-like tip that interacts with the immune cell receptor TIGIT and the CRC cell receptor Gal-GalNAc at two different sites.

## Results

### The Fap2-ECD forms a rod-shaped kinked β-helix

The genome of *F. nucleatum subsp. nucleatum* ATCC23726 contains 2111 ORFs[35], 21 of them containing autotransporter domains as annotated by InterProScan5[39]. These are on average larger proteins than other Fn ORFs, notably the largest two of them are the arginine-binding protein RadD and the virulence factor Fap2 (Fig. 1a)[35]. It has been shown that these two autotransporter adhesins are strongly expressed in Fn and localized in the OM[40]. Despite their importance in biofilm formation and attachment to the human host, no molecular characterization of either adhesin has been reported, probably limited by the lack of access to the purified proteins so far. As an obligate anaerobe and pathogen, cultivation and handling of Fn is restricted to biosafety level 2 laboratories with anaerobic cultivation conditions. This hampers the analysis under conditions that are required for many experiments, e.g., the co-culture with aerobic cell lines. As a prerequisite for these studies, we applied an *E. coli*-derived autotransporter system based on the AIDA-I adhesin, which facilitates diffuse adherence of enteropathogenic *E. coli*[41,42]. We used pAIDA1[43,44] as a template to produce Fap2 recombinantly on the OM surface of *E. coli*. Domain analysis indicated that the Fap2-ECD contains 3430 out of 3744 residues of the mature protein, followed by a C-terminal, 314 residues long autotransporter domain (Fig. 1b). To identify the regions of the ECD to be attached to the AIDA autotransporter, we predicted the structure of full Fap2 in a fragment-based approach (see "Methods" for details) using AlphaFold2[45]. This revealed a 47 nm long β-helix that comprises 2905 residues, where 325 residues close to the N-terminus form a region with low prediction confidence (Supplementary Fig. 1a). The β-helix is followed by a linker of 200 residues, which connects the β-helix to the autotransporter domain (Fig. 1c). While the sequence parts corresponding to the autotransporter, the linkers, and the β-helix are highly similar in Fap2 of two Fn strains, more variations are evident at the N-terminal part between Fap2 of Fn ATCC23726 and Fn ATCC25586 (Supplementary Fig. 1b).

The pAIDA1 plasmid contains the AIDA-I signal sequence, a tobacco etch virus (TEV) protease cleavage site, a Myc-tag and a ~5 kDa linker derived from the AIDA-I passenger domain upstream of the AIDA-I autotransporter[44,46]. Therefore, we omitted the Fap2 signal sequence and the proposed Fap2 linker between the β-helix and autotransporter (res. 3272–3471) and cloned only the variable N-terminal part and the entire β-helix (res. 42–3271) between the pAIDA signal sequence and the TEV site. To facilitate the isolation of pure Fap2-ECD, we introduced an octahistidine tag at its C-terminus and named the resulting plasmid pAIDA_TEV *fap2*EC (Fig. 1d, Supplementary Fig. 2a). We then expressed pAIDA_TEV *fap2*EC recombinantly in *E. coli* and indeed observed the presence of our fusion protein in the membrane (Supplementary Fig. 2b), indicating successful translocation of the Fap2-ECD through the AIDA autotransporter domain. After TEV cleavage, $Ni^{2+}$ affinity chromatography, and removal of remaining TEV protease by glutathione affinity chromatography, we obtained the 340 kDa large Fap2-ECD at >90% purity as qualitatively estimated by SDS-PAGE (Fig. 1e, Supplementary Fig. 2b–h), suitable for structural and biophysical analysis. In agreement with the β-helix of the Alpha-Fold2 prediction, the protein forms rods up to a length of about 50 nm long, as evident in cryo-EM micrographs (Fig. 1f).

Cryo-EM and SPA of the Fap2-ECD at an overall resolution of 4.7 Å (Supplementary Fig. 3a–d, 4a; Supplementary Table 1) revealed a total length of ~45 nm and width of 4.5 nm for the ECD. The individual strands of the β-helix are not entirely separated in the present density map, however a triangular-shaped groove of 2 nm width along the longitudinal axis is evident (Fig. 1g, Supplementary Movie 1). The β-helix domain is separated into two parts of 26 and 19 nm, respectively, by a kink of ~101° in the overall reconstruction. A comparison of different 2D class averages revealed variation in the kink angle: The majority of particles (51.5%) is found in 2D class averages with weak kinks, i.e., angles above 110°, whereas 38.2% are in strongly kinked classes with angles close to 90°. The remainder of particles are situated in top-view 2D class averages with indistinguishable angles (Supplementary Fig. 4b).

The 26 nm long, membrane-distal part has a thickening by 1.2 nm, which we term matchstick region. To improve the low local resolution in this region by eliminating heterogeneity through the variable kink angle, we carried out signal subtraction of the shorter β-helix part and local refinement of the tip of the longer, remaining part to 4.4 Å resolution (Supplementary Fig. 3e–h, 4a). With this, we identified a hook-like structure with two triangular extensions (Fig. 1g). These shapes in the cryo-EM density fit the lateral extensions at the membrane-distal tip of the β-helix in the AlphaFold model (residues K525–K836), and thus allowed rigid-body fitting of the predicted β-helical structure of the membrane-distal part (residues 290–1772) into the cryo-EM density. The structural model was then adjusted by a molecular dynamics (MD) simulation based fit in Namdinator[47], followed by manual model adjustment in Isolde[48] and real-space refinement in Phenix[49] (Fig. 1h). Although the final map-to-model FSC value was limited to 6.5 Å (Supplementary Fig. 3i), likely because of the non-separability of the β-strands in the β-helix, the data together with the model allowed interpretation of several unique key features of the structure. The map and the model reveal a unique hydrophobic longitudinal groove in the V-shaped β-helix that winds around the longitudinal axis of Fap2 (Fig. 1i, right panel). This groove is closed by the unstructured proline-rich, and amphiphilic N-terminus (residues N195–S361), which together result in a hydrophilic surface (Fig. 1i, left panel, Supplementary Fig. 3j, Supplementary Movie 1). The unstructured N-terminus is highly conserved in different fusobacterial species and in predicted autotransporter proteins of the *Pseudoleptotrichia goodfellowii* and *Leptotrichia* species that are known human pathogens (Supplementary Fig. 3k), also highlighting the conservation of the hydrophobic groove (residues 393–3271 in Fn ATCC23726) that is complemented by the N-terminus. No significant sequence similarity was identified outside the order *Fusobacteriales*, indicating that this particular structural motif is so far specific for the adhesins of these bacteria. The triangular extensions at the matchstick region result in two pits at the tip, which are present only in Fap2, but not in RadD (Supplementary Fig. 1c) and are probably involved in receptor binding.

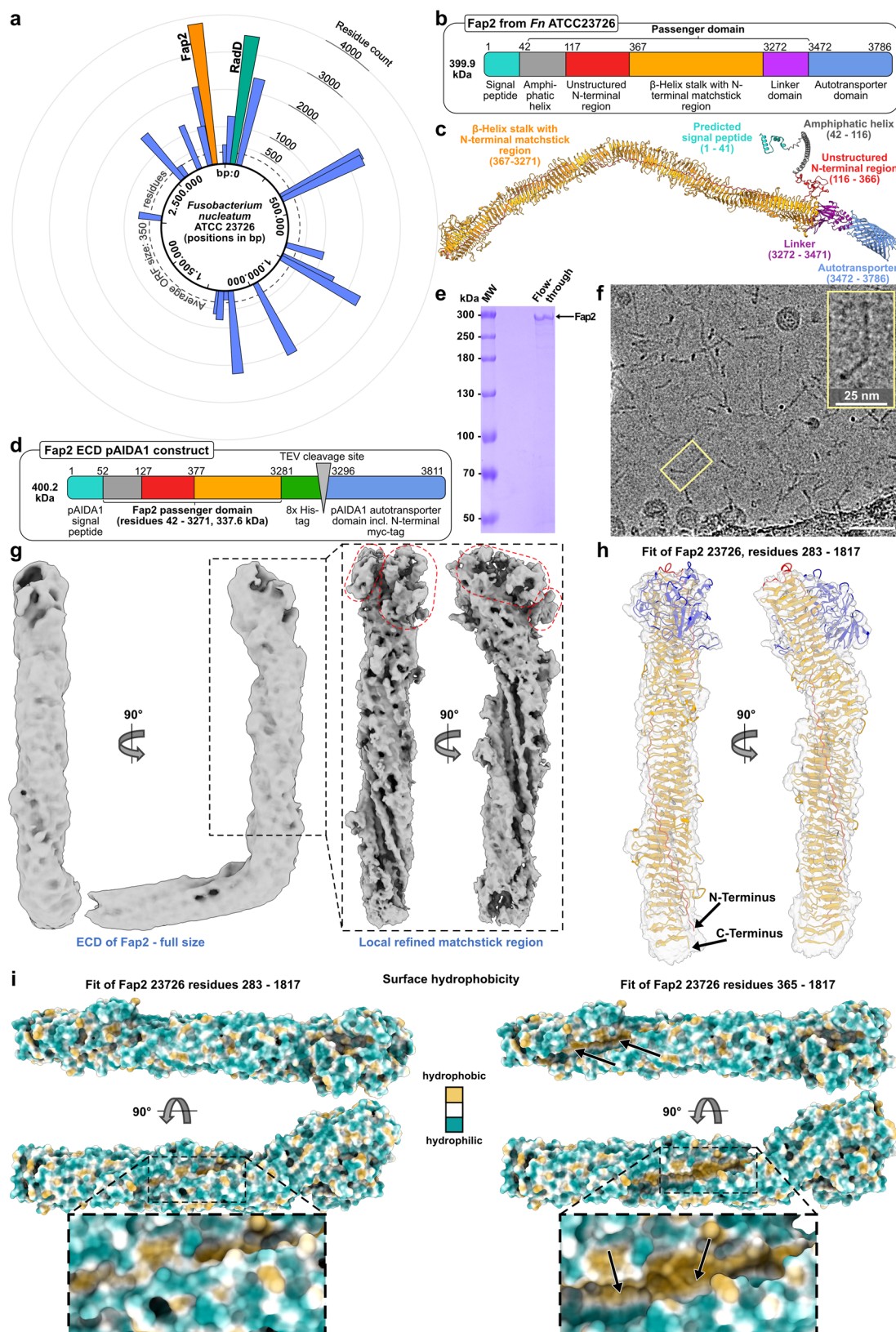

Our results provide a high-resolution structural analysis of Fap2 as a fusobacterial autotransporter adhesin and its putative receptor-binding β-helical ECD. It has an extended shape similar to other structurally described autotransporter adhesins, such as CdrA/B from *Pseudomonas aeruginosa*[50].

## Shape and size of recombinantly produced Fap2 resemble adhesins in the OM of Fn

To compare recombinantly purified Fap2 with native autotransporter adhesins in the Fn OM, we cultivated Fn ATCC25586 in the presence of the immortalized T-cell line Jurkat, which had been demonstrated

**Fig. 1 | Domain analysis, recombinant production and structure of Fap2.**
**a** Genome organization of Fn ATCC23726 with identified sequences for auto-
transporter adhesins (blue), in which Fap2 and RadD are indicated in orange and
green, respectively. **b** Domain organization of Fap2, as identified by Alpha-
Fold2. Note that the domains are not drawn to scale. **c** AlphaFold2 structure pre-
diction of Fap2 from Fn ATCC23726, colored by domain as in (**b**). **d** Schematic
overview of the pAIDA_TEV_Fap2EC expression plasmid, which contains the
unstructured N-terminal region, the matchstick and the β-helix of Fap2 (residues
42–3271). **e** SDS-PAGE, showing purified Fap2-ECD after release from the AIDA
autotransporter, Ni-NTA purification and removal of GST-TEV. MW molecular
weight marker. The image shows a representative Fap2 purification result out of ten
(see also Supplementary Fig. 2b, c). **f** Section of a Cryo-EM micrograph of purified

Fap2-ECD, recorded at 300 kV and 2.8 µm defocus. Scale bar: 50 nm. The image is a
representative of 12,663 micrographs. **g** Cryo-EM density maps of Fap2-ECD at 4.7 Å
resolution (left panel), and local refinement of the tip of the longer part, which
displays a hook-like end (right panel) with triangular extensions protruding from
the rod (indicated by red dashed lines). **h** Density map with residues 283–1817 of the
matchstick region and the β-helix of Fap2 fitted into the longer part. Residues
corresponding to extensions from the β-helix are colored blue, otherwise the color
code of panel **c** is used. **i** Surface representation of the longer part of the Fap2 ECD
model colored by hydrophobicity, in which the longitudinal groove is shown with
(left) and without (right) fitted unstructured N-terminus. Note the exposed
hydrophobicity without the N-terminus (right).

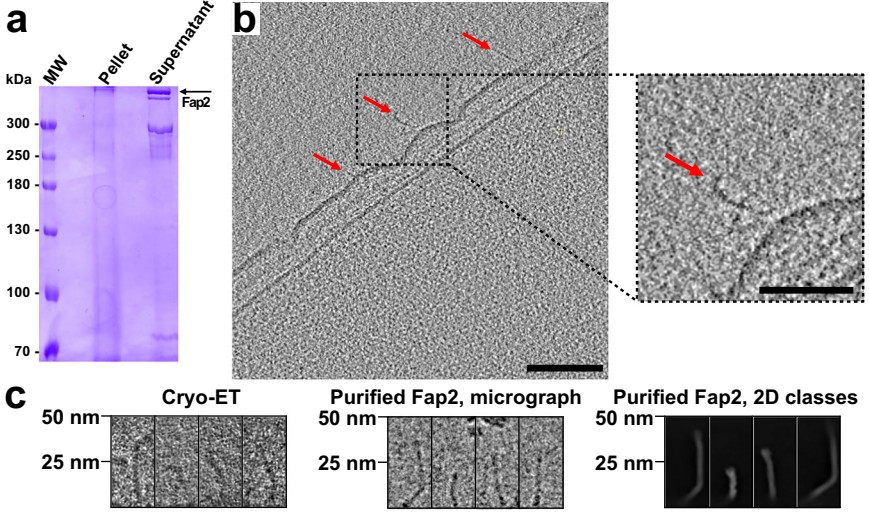

**Fig. 2 | Comparison of purified Fap2-ECD with native Fn outer membrane
adhesins. a** SDS-PAGE of the isolated OM fraction of Fn ATCC25586 after co-
cultivation with Jurkat cells. Proteins corresponding to the indicated bands were
identified as Fap2 and RadD in peptide fingerprint analysis (Supplementary Fig. 5a,
b). MW molecular weight marker. The experiment was repeated three times with
similar results. **b** Representative cryo tomogram out of 40 of a section of Fn
ATCC25586 including the inner membrane (IM), periplasm (PP) and outer mem-
brane (OM). Rod-like structures on the surface that represent autotransporter

adhesins in size and shape are indicated by arrowheads (scale bar: 100 nm). The
inset shows one of these magnified. Note that the ripples in the OM might be caused
by oxygen exposure of the anaerobic *F. nucleatum* before and during vitrification.
Scale bar: 100 nm, inset: 50 nm. **c** Selected rod-shaped OM surface protrusions
from Fn (left panel) detected in reconstructed tomograms (*n* = 40) in comparison
with different projection views of the recombinantly produced Fap2-ECD (middle
panel: micrograph, *n* = 12,663; right panel: 2D class averages, *n* = 81).

to increase Fap2 production in Fn ATCC23726[40]. We indeed observed
a strong expression of several >300 kDa proteins in the OM fraction
that make up more than 50% of all solubilized Fn OM proteins, as
evident in SDS-PAGE (Fig. 2a). Peptide fingerprint analysis of the
protein bands indicated that the three upper ones represent Fap2, as
reported for Fn ATCC23726 previously[40], while the lower band pre-
dominantly contains RadD (Supplementary Fig. 5a, b), both of them
being the largest ORFs in Fn[35,51] (Fig. 1a). To analyze the adhesins on
the native OM surface, we vitrified Fn ATCC25586 and visualized
their surfaces using cryo electron tomography (cryo-ET). In fact, we
could identify kinked rod-like structures protruding from the OM
(Fig. 2b). These cell attachments appear up to 50 nm long, which
matches the lengths of recombinantly produced Fap2-ECD as evident
in cryo-EM micrographs and 2D class averages (Fig. 2c). Moreover,
the membrane-distal ends of some cell attachments indicate thick-
enings, in agreement with the longer part of the Fap2-ECD (Fig. 1g,
Fig. 2b, c). After detergent extraction and isolation from the OM, the
cell attachments keep their shape and integrity, as indicated by SDS-
PAGE in which no degradation is evident, and by negative stain EM in
which rod-shaped kinked molecules of up to 50 nm length are visible
(Supplementary Fig. 5c–e). We however note that at the resolution
level of a cryo tomogram we cannot distinguish between Fap2 and
RadD because both appear to have the same length and a very similar
overall shape (Supplementary Fig. 1a).

Taken together, our findings confirm previous work that a large
number of Fap2 and RadD molecules are present in the Fn OM[40], and
different orientations alongside the membrane suggest that the Fap2
linker (res. 3272–3471) connects the Fap2-ECD in a flexible manner to
the autotransporter domain. Therefore, the entire flanks and tip of the
rod-like β-helix would be accessible for interaction with Gal-GalNAc
and TIGIT by Fap2.

To identify possible differences in post-translational modification
(PTM) of recombinantly produced Fap2-ECD with Fap2 in *F. nucleatum*,
we digested the proteins and compared the masses of the resulting
peptides. Mass spectrometry data revealed that the vast majority of
peptides are either unmodified or deamidated, whereas further mod-
ifications are of minor occurrence and similar in both Fap2 prepara-
tions (Supplementary Fig. 5f, Supplementary Table 2). The
unannotated mass shift of 60.98 Da in native Fap2 is compatible with
bound $Na^+$ and $K^+$, but no other known PTM (ABRF Delta Mass data
base). This indicates that recombinant and native Fap2 have no tract-
able differences in PTM that could influence their receptor binding
activities.

### Recombinantly produced Fap2 facilitates tumor cell attachment via Gal-GalNAc

Having verified successful recombinant production of Fap2 that
resulted in protein with native-like shape and domain arrangement, we

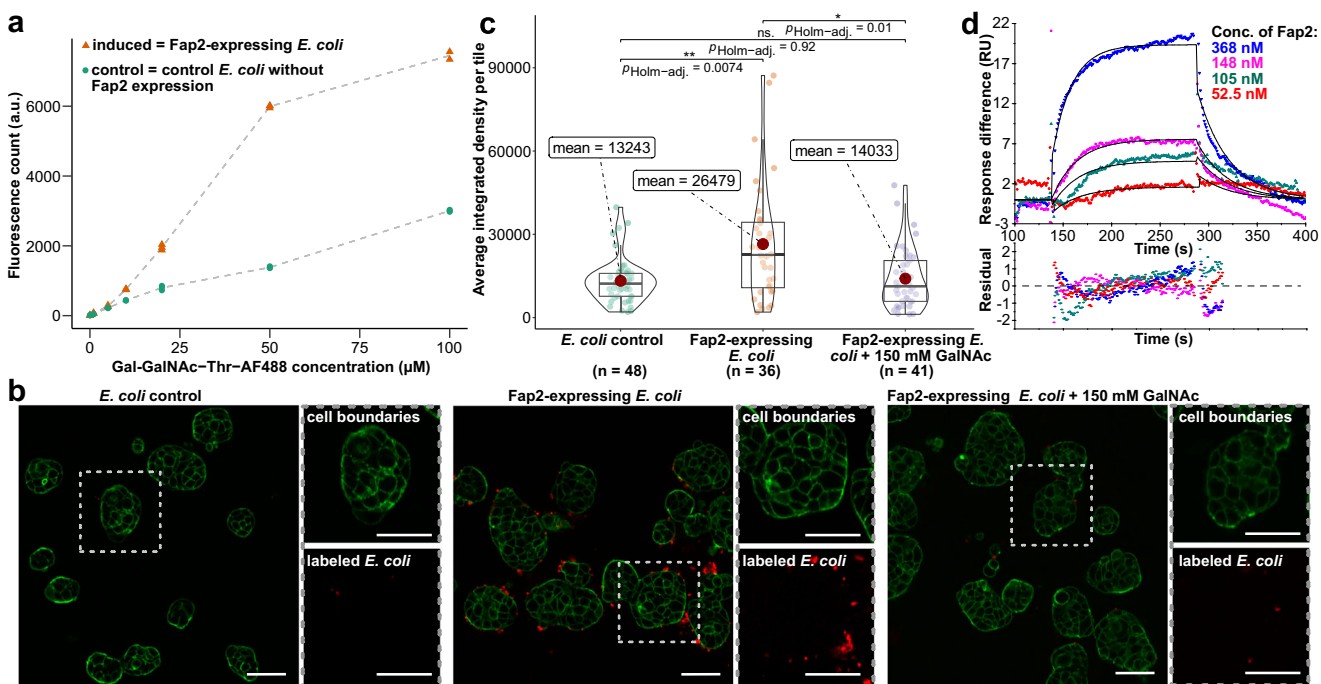

**Fig. 3 | Adhesion of Fap2 to Gal-GalNAc-Thr and hTIGIT. a** Concentration-dependent binding of fluorescently labeled Gal-GalNAc-Thr to Fap2-expressing *E. coli* and to control *E. coli* that express pAIDA without the Fap2-ECD. Data points represent individual measurements per concentration and dashed lines indicate the mean values. Data analysis and visualization was carried out in R using the ggplot2 package[83]. **b** Fluorescence micrographs that show cancer cell binding (HT-29) of Fap2-expressing *E. coli*. Less binding is observed with control *E. coli* (left panel). Cell binding is reduced in the presence of excess GalNAc (right panel). The images are representatives of data sets of 36, 48, and 41 micrographs for Fap2-expressing *E. coli*, control *E. coli*, and Fap2-expressing *E. coli* with GalNAc, respectively. Red: *E. coli* labeled with CellBrite Fix 555, green: Actin labeled with Phalloidin-iFluor 488. Scale bar: 50 μm. Note the unequal distribution of *E. coli* on HT-29 cells. **c** Quantification of experiments as in (**b**), with average integrated fluorescence signal of *E. coli* bound to HT-29. The number of micrographs (n) is indicated. The mean is shown as red dot and the boxes represent the interquartile range, with the median shown as horizontal line within the box. The distribution of data points is visually represented by violin plots. Statistical evaluation was done using the Games-Howell post-hoc test from the ggstatsplot package[82] in R. **d** Quantification of Fap2-ECD binding to hTIGIT-ECD using surface plasmon resonance. A global fit according to a 1:1 binding model (black solid lines) reveals a $K_D$ of ~0.6 μM (Supplementary Table 3). Lower panel: residuals of fit.

set out to test the functionality of the recombinant Fap2-ECD, i.e., the association with its two known receptors, Gal-GalNAc and TIGIT. The disaccharide Gal-GalNAc is an abundant O-glycosylation of surface proteins of colon cancer cells, thereby serving as a biomarker for CRC[37], and facilitates tumor colonization of Fn in orthotopic mouse models[36]. To examine whether recombinant Fap2 on the *E. coli* surface binds to Gal-GalNAc, we transformed the bacteria with pAIDA_-TEV *fap2*EC, induced expression, and incubated with a fluorescently labeled Gal-GalNAc-Thr adduct (Supplementary Fig. 6a–c). In fact, the difference between Gal-GalNAc binding for Fap2-expressing *E. coli* and *E. coli* transformed with pAIDA without Fap2 increased until a plateau at Gal-GalNAc concentrations of 50–100 μM is reached (Fig. 3a). This means that recombinant Fap2 binds to the cancer cell receptor at least as well as native Fap2 on Fn, where cellular and tissue binding studies were carried out with Gal-GalNAc concentrations in the mM range[36].

Next, we tested whether the Fap2-expressing *E. coli* are able to adhere to colon cancer cells. For this, we stained *E. coli* with a membrane-specific dye and used fluorescence microscopy to study their attachment to the CRC cell line HT-29. Indeed, only Fap2-expressing *E. coli* attached to the surface of HT-29 in small clusters, whereas *E. coli* did not (Fig. 3b, c, Supplementary Fig. 6d). Notably, the bacteria are not evenly distributed among cells, but cluster at higher numbers on some, whereas other cells are practically unoccupied. Similar observations were made for Caco-2 CRC cells (Supplementary Fig. 6e). These data indicate either uneven distribution of Gal-GalNAc on different CRC cells in the population, different surface accessibilities of the glycan to the bacteria, or auto-aggregation of Fap2-expressing *E. coli* prior to cell binding. Our experiments suggest that

recombinant Fap2 provides sufficient affinity to allow association of Fap2-expressing *E. coli* to cancer cells, even to those with low Gal-GalNAc surface levels such as HT-29[36], and that no other *E. coli* surface proteins bind to a surface receptor on the CRC cells with an affinity comparable to that of Fap2 for Gal-GalNAc. We note, however, that direct binding of Fap2 to Gal-GalNAc cannot be demonstrated without fluorescence co-localization experiments.

To verify that Fap2-expressing *E. coli* attach to Gal-GalNAc and not to further possible receptors on cancer cells, we carried out a competition assay with GalNAc in solution. As expected, attachment to HT-29 was tremendously reduced in the presence of 150 mM soluble GalNAc almost to the level of non-induced bacteria (Fig. 3b, c). We therefore conclude that the interaction is specific between Gal-GalNAc and Fap2, involving no other receptors of *E. coli* and HT-29 cells.

Together, our results demonstrate that Fap2 attached to the *E. coli* autotransporter AIDA is functional and allows binding to CRC cells. Consequently, we present a simplified cell-based system that allows to assess Fn cell adhesion in its natural context without the need for anaerobic work or higher biosafety levels, enabling co-culture experiments with oxygen-dependent human cell lines and clinically relevant experiments to a broader research community.

## Recombinantly produced Fap2 interacts with TIGIT

After confirming the functional binding of recombinant Fap2 to cancer cells via the glycan receptor Gal-GalNAc, we next addressed its association to the receptor on immune cells, TIGIT[33]. To study the interaction of Fap2 with native-like, human TIGIT (hTIGIT) that has two N-glycans on N32 and N101[52], we expressed the ECD of hTIGIT

(residues 1–141; hTIGIT-ECD) together with a C-terminal eGFP and StrepII tag in Expi cells and purified the protein (Supplementary Fig. 7a–c). The blurred band of purified hTIGIT-ECD on SDS-PAGE indicates that the protein is indeed glycosylated, likely at both predicted sites, and we proceeded to test the interaction of purified hTIGIT-ECD with Fap2.

We then carried out a qualitative pulldown assay for which we immobilized Fap2-ECD on NiNTA and loaded secreted hTIGIT as prey. Indeed, we identified hTIGIT-ECD as binding partner (Supplementary Fig. 7d), whereas non-glycosylated hTIGIT expressed in and purified from *E. coli*[53] did not bind to Fap2 (Supplementary Fig. 7e, f). This indicates the direct or indirect involvement of one or both glycans in binding, similar to the reported interaction of hTIGIT with the poliovirus receptor (PVR)[52]. To evaluate the physiological relevance of the Fap2-hTIGIT interaction in comparison to other known hTIGIT interaction partners (PVR and Nectin-2), we set out to quantify the affinity of glycosylated hTGIT-ECD and Fap2-ECD using surface plasmon resonance (SPR). The obtained dissociation constant ($K_D$) of ~0.6 μM (Chi² = 0.477, $k_{on}$ = 52,900 ± 3120 $M^{-1}s^1$, $k_{off}$ = 0.03 ± 0.001 $s^{-1}$; Fig. 3d, Supplementary Table 3, Supplementary Fig. 7g) is between those reported for hTIGIT and PVR of 1–3 nM[54,55] and for hTIGIT and Nectin-2 of 6 μM[56]. Given the fact that NK cells are deactivated via hTIGIT interaction to PVR[57], but also via interaction with the low-affinity receptor Nectin-2, both of them strongly expressed on the surface of cancer cells[58], we conclude that the Fap2 affinity to hTIGIT is sufficient for NK cell deactivation, independent of cancer cell receptor binding to hTIGIT.

### hTIGIT binds to the membrane-distal tip of Fap2

To identify where and how Fap2 binds to hTIGIT, we vitrified the complex of Fap2-ECD and hTIGIT-ECD after pulldown (Supplementary Fig. 7d, Supplementary Fig. 8a) for cryo-EM and SPA. We found that at the tip of the matchstick, density appeared fragmented likely because of a mixture of Fap2 with or without bound hTIGIT. We therefore proceeded with focused 3D classification of the tip and then combined subsets with clearly defined density beyond the Fap2 matchstick, resulting in 45,207 out of 73,482 particles for final refinement (Supplementary Fig. 8e, f). The resulting density map with a global resolution of 6.0 Å (Supplementary Fig. 8) revealed a long, kinked, rod-like structure, similar to the Fap2-ECD alone (Fig. 4a). In the complex, the tip of the longer part of the β-helical rod has an additional, angular-shaped density, whereas the tip of Fap2 alone appears round-shaped (Fig. 4b). After density map subtraction of Fap2 alone, a cuboid shaped density of 5 nm length and 3 nm width at the topmost part of the Fap2 β-helix remained, which is large enough to accommodate one hTIGIT-ECD (Fig. 4c, Supplementary Fig. 8g, Supplementary Movie 2). No other difference densities are evident outside of the Fap2 structure, indicating that hTIGIT binds only at the tip and that binding does not induce structural changes elsewhere in Fap2 that are apparent at this level of resolution (Supplementary Fig. 8g).

Due to the limited resolution of the cryo-EM map, we applied an integrative approach to model the hTIGIT-ECD (residues 23–128) on Fap2. For this, we first identified residues of Fap2 that are in close spatial proximity to the additional density in the complex and then docked the polypeptide part of hTIGIT-ECD (PDB 3UCR) to this Fap2 part using HADDOCK[59] (Supplementary Table 4). The best cluster of docking poses (score: −14.6 ± 6.7) revealed that loop P87-G92 on hTIGIT-ECD intercalates into a cleft at the Fap2 tip that is formed between D360, T397, R398, E425, N494, D496, and W540 (Fig. 4c, d). We then tested the docked models in MD simulations. In 7 out of 10 replicates, hTIGIT remained bound throughout the 150-ns trajectories (Supplementary Movie 3). Fap2-W540 forms a hydrogen bond with hTIGIT-P65 and simultaneously interacts via hydrophobic contacts with P67, which intercalates deeply into the cleft at the Fap2 tip. Hydrogen bonds between Fap2-W540, R427, S371, K368 and R398, and

hTIGIT-Q22, H24 A30, V32 and P65 are identified as the major contributors to the interaction (Fig. 4d, Supplementary Table 5). Fap2-R427 forms a hydrogen bond with the hTIGIT-Q22 and H24 side chains. Fap2-N541, S371 and K368 form a C-shaped interaction interface and participate in consistent hydrogen bonds with hTIGIT-W53, A30 and V32, respectively, throughout the MD simulation replicates in which the complex remained intact (Supplementary Movie 3).

We next created a Fap2-ECD mutant of the binding interface that would remove the identified main-chain and side-chain hydrogen bonds (K368P_S371A_R427A_W540A), expressed it on the *E. coli* surface and tested binding of the bacteria to HEK293 cells that overexpress hTIGIT-eGFP. Unexpectedly no significant differences in hTIGIT-expressing cell binding between *E. coli* that express Fap2-ECD wild type (WT) and this mutant became evident (Fig. 4e, f), with nearly identical mean and average binding values. Furthermore, in a pulldown assay using purified Fap2 WT and the mutant, no qualitative differences in their ability to bind hTIGIT-ECD were observed (Supplementary Fig. 9a). These results are an indication for further possible binding modes of the hTIGIT-ECD loop within the cleft at the Fap2 tip, or redundancy provided by other Fap2 side chains.

The Fap2/hTIGIT interface is on the same side as the hTIGIT dimerization interface, as evident in the co-structure of hTIGIT/PVR (PBD 3UDW), but differs considerably with respect to orientation and surface accessibility (Supplementary Fig. 9b). Both glycosylation sites (N32 and N101) protrude away from Fap2 in the complex, however, N32 is in close spatial proximity to a predominantly hydrophilic gap between Fap2 and hTIGIT (Supplementary Fig. 9c). This suggests an indirect role for the hTIGIT glycans in interacting with Fap2, probably by introducing slight structural changes in the intercalating loop.

In summary, our structural assessment of the Fap2-ECD/hTIGIT-ECD complex reveals binding of hTIGIT to Fap2 at the maximum distance from the Fn surface. Whereas docking and MD simulation suggest a particular mode of interaction via a hTIGIT loop that intercalates into a cleft at Fap2, site directed mutagenesis did not confirm the binding model. Therefore, the docking and MD simulation results have to be interpreted with caution and further structural analysis of the complex at near-atomic resolution is required to understand the binding mechanism of Fap2 to hTIGIT at its full extent.

### Gal-GalNAc binds into a pocket at the membrane-proximal side of the matchstick tip

To gain insight into the binding site of the CRC cell receptor Gal-GalNAc and whether both receptors can bind to Fap2 simultaneously, we docked the Gal-GalNAc disaccharide attached to Thr, as used in our binding experiment (Fig. 3a) and as present in most O-glycosylations in the colon on mucin-2[60], to Fap2-ECD. Half of the docking poses, including the best one (score of ‑6.1 kcal/mol), were placed in a pit that is located at the back side of the Fap2 matchstick region, opposite to the hTIGIT binding site, facing away from the tip (Fig. 5a, Supplementary Fig. 10a). Notably, this pit shows two openings that would enable the entry and exit of the polypeptide part of O-glycosylated receptors. The Thr attached to Gal-GalNAc in the obtained docking pose aligns with the openings (Fig. 5a), which would be consistent with Fap2 hanging on a Gal-GalNAc-glycosylated polypeptide like a hook on a chain. To identify whether more Gal-GalNAc binding sites exist on the Fap2 β-helix, we carried out further dockings of a larger area of the Fap2-ECD, including the entire matchstick region and the largest part of the longer β-helix part ahead of the kink. Moreover, we docked Gal-GalNAc-Thr to the analogous region of the predicted RadD model (Supplementary Fig. 1a). In Fap2, the highest scores consistently resulted from poses that were docked into the same pit on the matchstick in a similar orientation (Supplementary Fig. 10b). In contrast, docking to RadD resulted in a high variance of different poses (Supplementary Fig. 10c). This shows that the Gal-GalNAc binding pit is specific for Fap2, and is consistent with the absence of such a pit in the

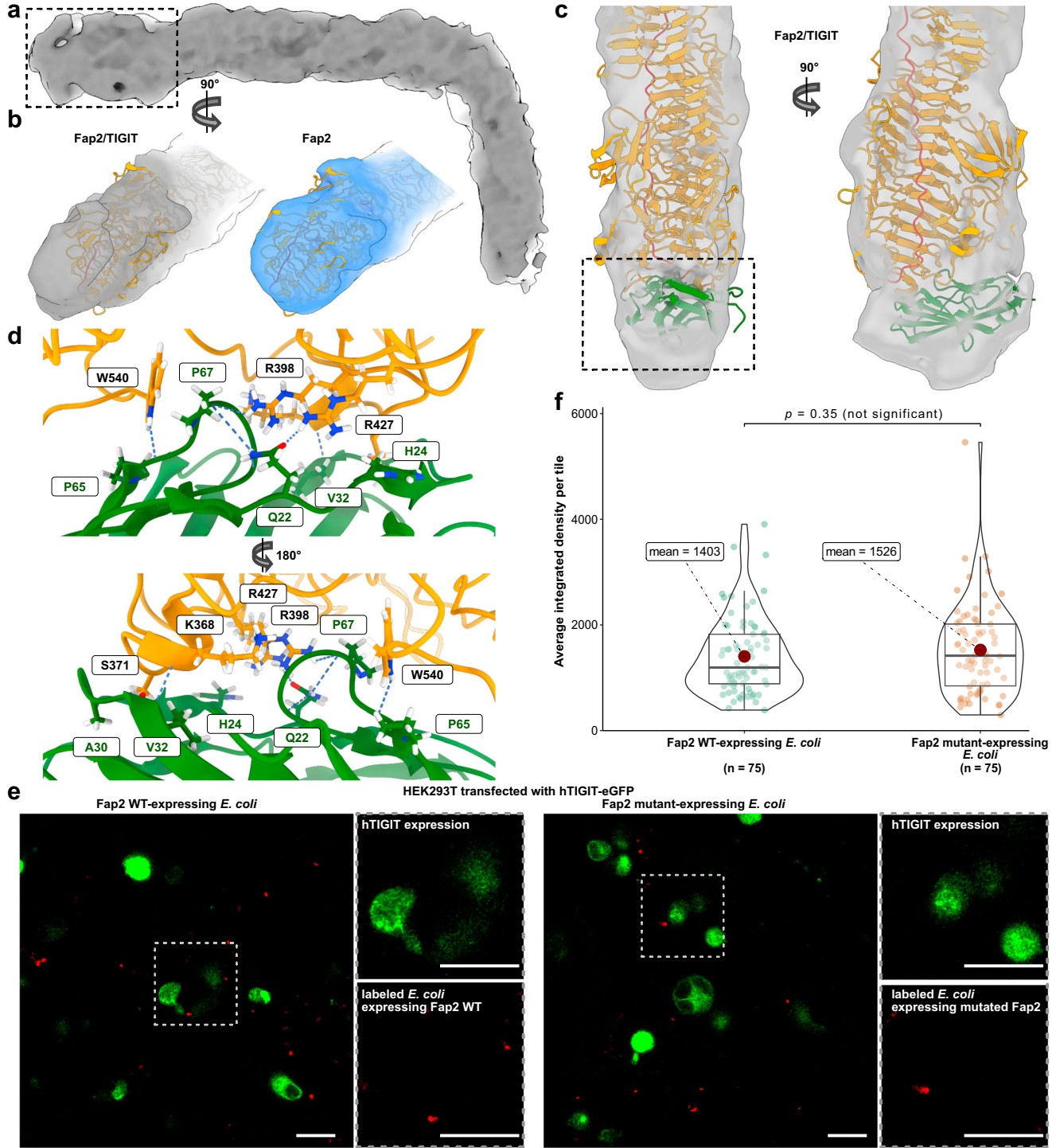

**Fig. 4 | Structure of the complex of Fap2-ECD and hTIGIT-ECD. a** Cryo-EM density map of Fap2-ECD/hTIGIT-ECD at 6.0 Å resolution. **b** Comparison of the cryo-EM densities of the membrane-distal tip regions of Fap2-ECD/hTIGIT-ECD (left) and Fap2-ECD (right), as indicated by the box in (**a**). The Fap2 model is shown in orange for both maps. **c** Docking of hTIGIT-ECD (green) to Fap2-ECD (orange) with HADDOCK. The best docking pose is shown and is in agreement with the cryo-EM density. **d** Proposed interaction site of Fap2 and hTIGIT as indicated by the box in (**c**). The loop G66-P67-G68 of hTIGIT intercalates into a gap in Fap2, and main chain and side chain interactions as identified in docking and verified by MD simulations are highlighted. **e** Fluorescence micrographs that show HEK293 cells transfected with hTIGIT-eGFP and binding of Fap2-expressing *E. coli* (WT and

mutant K368P_S371A_R427A_W540A). Red: *E. coli* labeled with CellBrite Fix 555, green: hTIGIT-eGFP. Scale bar: 50 µm. The images are representatives of data sets of 75 micrographs each. **f** Quantification of experiments in (**e**), with average integrated fluorescence signal of *E. coli* bound to HEK293T transfected with hTIGIT-eGFP determined from 75 micrographs each. The integrated fluorescence of *E. coli* was detected with the abundance of hTIGIT-eGFP signal as a prerequisite. The mean is shown as red dot and the boxes represent the interquartile range, with the median shown as horizontal line within the box. The distribution of data points is visually represented by violin plots. Statistical evaluation was done using the *t*-test from the ggsignif[105] package and visualized with ggstatplot[82] in R.

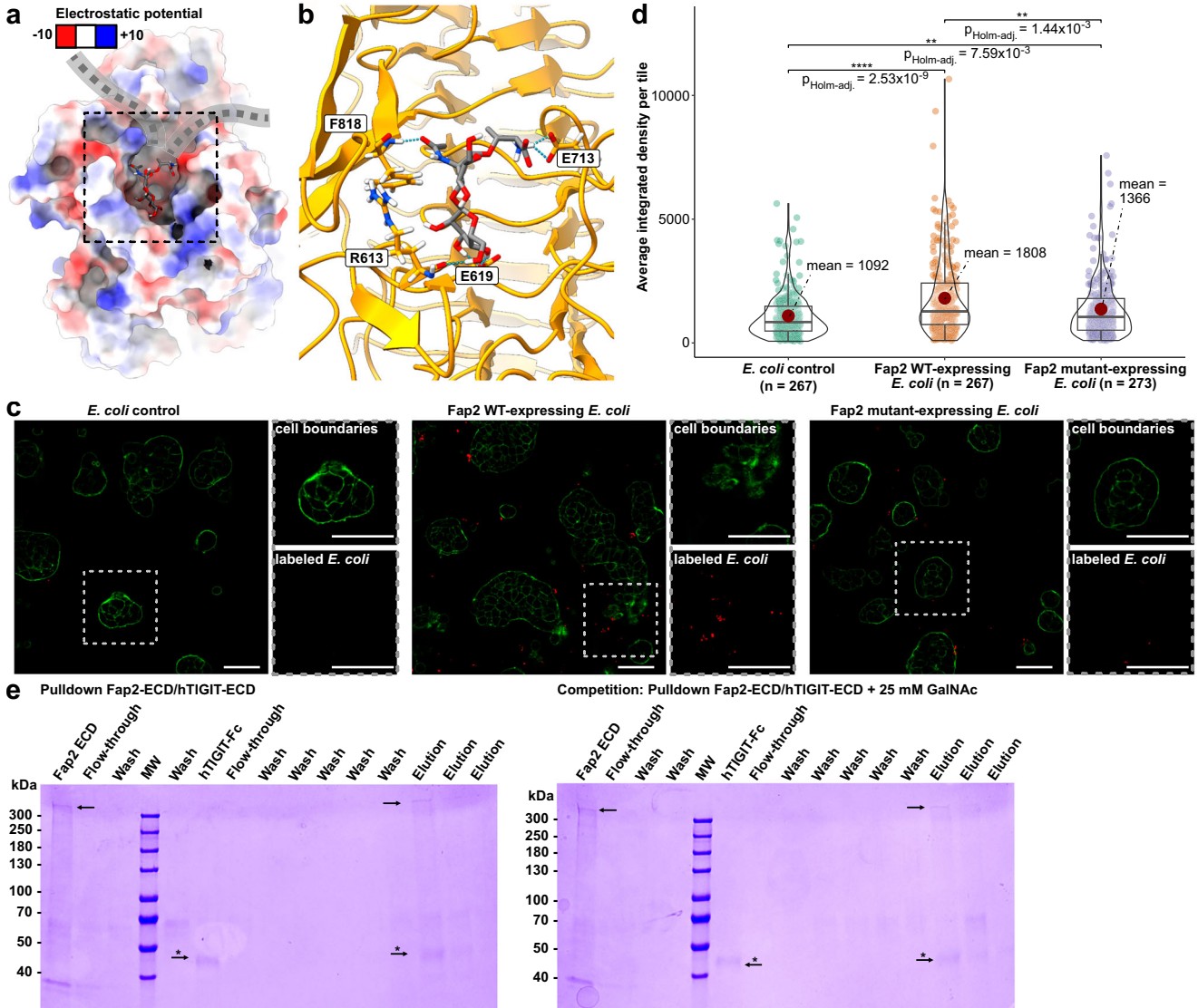

**Fig. 5 | Docking of Gal-GalNAc-Thr to Fap2-ECD and validation of binding site.**
**a** Surface representation of Fap2 colored according to Coulomb potential (red: negative, blue: positive charges) with docked Gal-GalNAc-Thr. A representative frame of the MD simulation of the best docking pose is shown (Supplementary Movie 4). The Fap2 model has two clefts adjacent to the docking sites, which is in agreement with the polypeptide chain of the Gal-GalNAc-containing O-glycosylated receptor going through (indicated by gray dashed line). **b** Hydrogen bonds between the docked Gal-GalNAc-Thr and R613, E619, E713, and F818 of Fap2, which remain most consistent in MD simulation. **c** Representative fluorescence micrographs that show HT-29 cell binding of Fap2-expressing *E. coli* (analogous as in Fig. 3b), in which Fap2 WT and Fap2 mutant R613P_E619A_F818P are compared. Red: *E. coli* labeled with CellBrite Fix 555, green: Actin labeled with Phalloidin-iFluor 488.

Scale bar: 50 μm. The images are representatives of data sets of 267 micrographs each for Fap2 WT-expressing *E. coli* and control *E. coli*, and 273 micrographs for Fap2 mutant-expressing *E. coli*. **d** Quantification of experiments in (**c**), with average fluorescence signal of *E. coli* bound to HT-29 determined from the indicated number of micrographs each. The mean is shown as red dot. The boxes represent the interquartile range, with the median shown as horizontal line within the box. The distribution of data points is visually represented by violin plots. Statistical evaluation was done using the Games-Howell post-hoc test from the ggstatsplot package[82] in R. **e** Pulldown of Fap2-ECD with hTIGIT-ECD in the absence (left; $n = 3$) and presence (right; $n = 1$) of 25 mM GalNAc. The arrow indicates the position of Fap2-ECD, the arrow with asterisk indicates the position of hTIGIT-ECD-Fc.

predicted model of the otherwise structurally very similar RadD (Supplementary Fig. 1c).

To confirm the relevance of docking and analyze the role of individual H-bond interactions, we carried out MD simulations of the docked Gal-GalNAc-Thr-Fap2 complex. Gal-GalNAc-Thr remained bound and within the pit in 5 out of 9 replicates (Supplementary Movie 4), adopting slightly varying conformations. Hydrogen bond analysis between Fap2 and Gal-GalNAc-Thr of trajectories where the ligand remained bound revealed Fap2-F818, E619 and E713 as major contributors to ligand binding (Supplementary Movie 4). E713 undergoes H-bonds with the Thr amino group of Gal-GalNAc-Thr and not with the glycan moiety, therefore this interaction might differ largely

in a Gal-GalNAc-containing polypeptide. The most stable conformation observed showed GalNAc O5 interacting with the F818 backbone amino group. The backbone carbonyl group of R613 interacts with Gal O8; and E619, located deep inside the binding pit, consistently forms hydrogen bonds with Gal O5/O6, which appears to be a prerequisite for stable glycan binding (Fig. 5b, Supplementary Table 6).

To confirm the key roles of these residues in the pocket for Gal-GalNAc binding, we created the Fap2 triple mutant E619A_R613P_F818P (Ala mutation for side-chain interacting E619, Pro mutations for main-chain interacting R613 and F818), and tested cancer cell adhesion of *E. coli* that express the resulting Fap2-ECD variant. The adhesion of *E. coli* that express Fap2-ECD_E619A_R613P_F818P was indeed slightly

impaired, with an average binding signal reduction to ~70% of the wild type (Fig. 5c, d). This difference is significant, but not as pronounced as the reduction in binding caused by the addition of an excess of free GalNAc (Fig. 3b). We therefore conclude that these residues and their proposed H-bonds are important for binding, albeit other residues might provide redundancy. Further high-resolution structural research is required to confirm the exact molecular nature of the interaction.

The hTIGIT interaction site of Fap2 is ~3 nm away from the Gal-GalNAc binding pit, which indicates that both receptors have their unique and independent binding sites. We further predicted binding sites on the N-terminal matchstick region using the neural network DeepSite[61], which resulted in the identification of our binding sites for both hTIGIT and Gal-GalNAc with the highest prediction scores (Supplementary Fig. 10d). Gal-GalNAc is present as O-glycosylation on cancer cells in the colon[37], and the Fap2 immune cell receptor hTIGIT has two N-glycosylation sites that are crucial for its interaction with PVR[52]. To assess whether the attached glycan receptor inhibits Fap2 binding to hTIGIT despite the physical separation of the binding sites, we carried out a pulldown assay of Fap2-ECD with hTIGIT-ECD in the presence of 25 mM GalNAc, as used previously in hemagglutination assays[36] (i.e., a ~$10^3$ fold molar excess over hTIGIT-ECD). We found that Fap2-ECD still bound and pulled down hTIGIT-ECD in the presence of GalNAc, qualitatively without any difference in the band intensities of hTIGIT-ECD and Fap2-ECD (Fig. 5e). This indicates that co-binding of both receptors to Fap2 is generally possible due to physical separation of the two binding pits.

In conclusion, the docking and MD simulation of Gal-GalNAc-bound Fap2-ECD together with site-directed mutagenesis reveal not only mechanistic details of the Fn interaction with cancer cells, but also expose the physical separation of two interaction sites on Fap2 for the receptors of different cell types.

## Discussion

Here, we report the recombinant production and the functional and structural analysis of the infection- and cancer-relevant fusobacterial autotransporter adhesin Fap2. The β-helical part of the ECD that is relevant for host cell binding is already 338 kDa large, posing a great challenge for production in *E. coli*, which is by far the most widely used and established protein expression system. Only very few proteins above 2500 residues are reported in the protein data bank (PDB) with *E. coli* as expression system, e.g., Tc toxins[62], Talin1[63], *M. tuberculosis* fatty acid synthase[64] and *Clostridioides difficile* Toxin A[65]. Previous work used the AIDA-I-based and other autodisplay systems for cell surface display of a variety of proteins and applications[43,44,66–68], including cleavable versions[44,68,69]. Fn Fap2-ECD in this work is, to our knowledge, the first membrane-associated protein of such a large size that has been produced in *E. coli* using the AIDA-I autodisplay system, thereby underlining the broad applicability thereof.

Binding experiments to Gal-GalNAc-Thr and hTIGIT proved functionality of recombinantly produced Fap2-ECD. Using indirect binding competition experiments, we show that our recombinantly produced Fap2-ECD exhibits affinity to Gal-GalNAc-Thr that is at least as high as described for native Fap2 on Fn. For Fap2-ECD and hTIGIT, our measured $K_D$ of ~0.6 μM is the first reported quantitative value of this interaction. When compared to the affinities of other autotransporter adhesins with their receptors, e.g., *Haemophilus influenzae* autotransporter lipoprotein P4 with fibronectin and laminin ($K_D$ of about 10 nM)[70], *E. coli* UpaB with fibronectin ($K_D$ of 45 nM)[71], or *Yersinia enterocolitica* YadA with collagen ($K_D$ of 170 nM)[72], the Fap2/hTIGIT affinity appears weak. This is not unexpected when considering the binding of Fap2 to hTIGIT in the context of functional interaction with NK cells and tumor cells. The pathophysiological purpose of the Fap2/hTIGIT interaction is to deactivate NK cells and effector T-cells by inhibitory signaling through hTIGIT activation after Fap2 binding[73]. Thereby, transient interaction is likely sufficient to initiate TIGIT

signaling in immune cells, as also suggested by the fact that the even weaker interaction of Nectin-2 with hTIGIT ($K_D$ of ~6 μM)[56] is enough for this purpose[58]. On the other hand, the association of Fap2 to Gal-GalNAc allows colonization of tumors[22,36] and placenta[74], which typically requires high affinity. However, individual protein-glycan interactions are typically considered to be weak[75], and data from us (Fig. 3a) and others[36] suggest the same for Fap2 and Gal-GalNAc. Therefore, multivalent interactions are likely to be required. The high density of Gal-GalNAc on the surface of tumor cells and the high level of Fap2 expression in Fn[40] provide the basis for multivalency. Notably, although our data indicate different binding sites for the two Fap2 receptors, the possibility of interference between their binding cannot be ruled out due to a potential involvement of hTIGIT glycans and/or conformational changes. Thus, a comparably weak Fap2 affinity to hTIGIT might have an evolutionary advantage when the transient interaction, which is sufficient to deactivate the NK cell activity, is rapidly released and the same Fap2 molecule becomes available for interaction with tumor cells.

Using molecular docking and MD simulations, we identified the Gal-GalNAc binding site of Fap2, and site-directed mutagenesis supports our proposed binding model. In silico analysis of Fap2/Gal-Gal-NAc binding indicates the possibility for ambiguity of the binding pit towards further glycans. This finding is not surprising, as previous studies have demonstrated that the binding and invasion of CRC cells by Fn is affected by the presence of galactose-containing sugars[76].

Our model of the Fap2-hTIGIT interaction derived by a low-resolution complex structure, docking and MD simulations includes, among multiple hydrogen bonds, a stacking-like interaction between a Pro and Trp residue which has been found to be stable and to play an important role in protein-protein interactions[77,78]. However, as site-directed mutagenesis of the proposed binding site of Fap2 did not exhibit a significant decrease in affinity, the hydrogen bonds formed by the involved residues are possibly redundantly replaced by others. At this point, our integrated approach reaches its limit and high-resolution data of the complex are required to ultimately visualize the side-chain interactions involved at the binding site.

Taken together, our results suggest a unique mechanism of Fap2 interaction with both receptors, that takes the long shape of Fap2 and the hTIGIT binding site at its membrane-distal end into account. We propose that Fn applies Fap2 as a molecular spear to deactivate tumor-invading immune cells already at the maximum possible distance through transient hTIGIT interaction and subsequent signaling (Fig. 6). This is achieved through the ~45 nm long rod formed by the Fap2 β-helix that acts as a spacer to keep active NK cells at a safe distance from Fn and invaded tumors. At the same time, Fn is tightly associated to CRC cells in tumors via multiple Fap2/Gal-GalNAc interactions at the back side of the Fap2 tip, analogous to grappling hooks. This is enabled by high levels of Fap2 found on the Fn surface, as evident in Fig. 2a. Specific inhibition of either of these Fap2-mediated interactions is likely to have a tremendous impact on Fn tumor colonization and is thus a promising future strategy to combat CRC, particularly metastasis and chemotherapeutic resistance.

## Method

### Source of plasmids and genes for design of expression constructs

The *E. coli* expression vector pAIDA1 was a gift from Gen Larsson (Addgene plasmid #79180; http://n2t.net/addgene:79180; RRID:Addgene_79180)[44]. The *fap2* gene sequence from *Fusobacterium nucleatum subsp. nucleatum* ATCC23726 (FusoPortal[35], Gene 2068) was codon-optimized for expression in *E. coli* and purchased from GenScript. Oligonucleotides for cloning were purchased from Sigma-Aldrich. Custom-synthesized DNA corresponding to the hTIGIT gene sequence (Uniprot ID: Q495A1) was purchased from GenScript and cloned in a modified pEG-BacMam vector with C-terminal eGFP[79].

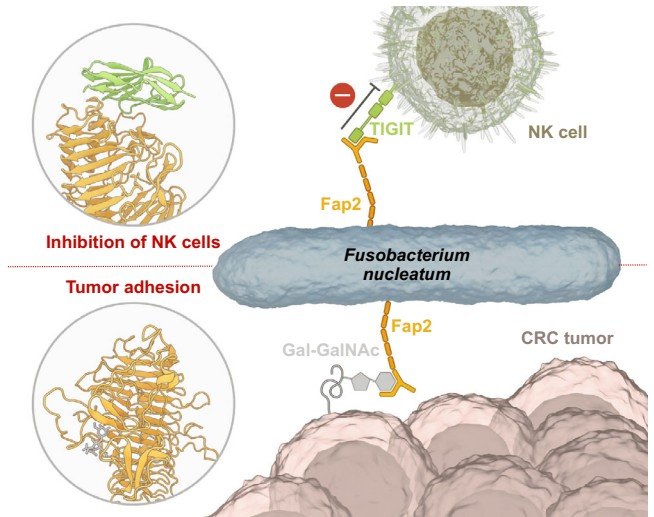

**Fig. 6 | Proposed mechanism.** Proposed mechanism of how Fn applies Fap2 to associate to tumors and deactivate tumor-invading NK cells through interaction with Gal-GalNAc and hTIGIT on the back side and front side of the Fap2 matchstick tip, respectively. Thereby, Fap2 simultaneously acts as a molecular spear that deactivates NK cells via TIGIT interaction and as a molecular grappling hook that anchors the bacteria to the tumor. Artwork created by Barth van Rossum (Leibniz FMP).

pGEX4T3-GST-TEV was a gift from Robert Sobol (Addgene plasmid #177142; http://n2t.net/addgene:177142; RRID:Addgene_177142)[80] and was modified to replace the thrombin cleavage site by an HRV3C cleavage site.

## Cloning of expression plasmids for Fap2 and hTIGIT

The synthetic gene coding for the Fap2-ECD (residues 42–3271) from *Fusobacterium nucleatum subsp. nucleatum* ATCC23726 was cloned into the pAIDA1 backbone in frame with the pAIDA1 signal peptide. An 8x His-tag was cloned between the C-terminal end of Fap2-ECD and the TEV cleavage site in pAIDA1 with 3 amino acid linkers before (GSA) and after (SAG) the His-tag, resulting in pAIDA_TEV_*fap2*EC (Fig. 1d). The GalGalNAc mutant (E619A_R613P_F818P) of pAIDA_TEV_*fap2*EC was generated by introducing single mutations sequentially using the KLD Enzyme Mix (New England Biolabs, #M0554S), whereas the hTIGIT mutant (K368P_S371A_R427A_W540A) of pAIDA_TEV_*fap2*EC was created by incorporation of a DNA segment containing the desired mutations.

The gene coding for the ECD of hTIGIT including the signal peptide (residues 1–141) was cloned into the pEG BacMam backbone downstream of the CMV promoter in fusion with an HRV3C site and a C-terminal eGFP with a GGS linker before eGFP. A Twin-Strep-tag was attached to the C-terminus of eGFP after a 5 amino acid linker (SGRMA). To generate pcDNA3.4_*hTIGIT(1-162)_eGFP* for cell binding assays, the gene coding for the ECD of hTIGIT including the signal peptide and transmembrane region (residues 1–162) was cloned into the pcDNA3.4 vector downstream of the CMV promoter in fusion with eGFP, followed by a Twin-Strep-tag. A GGS linker was attached to the C-terminal end of hTIGIT and an SGRMA linker was inserted at the C-terminus of eGFP. To generate pET19_*hTIGIT*_EC, the gene coding for the ECD of hTIGIT (residues 23–128) was cloned into pET19 downstream of the T7 promoter. A 6x His-tag followed by a SSG linker before an HRV3C protease cleavage site was attached at the N-terminal side of the hTIGIT ECD.

All gene insertions were carried out using the NEBuilder HiFi DNA Assembly kit (New England Biolabs, #E5520). Supplementary Table 7 shows an overview of all oligonucleotides used in this work.

## Expression and purification of Fap2-ECD

BL21(DE3) Star *E. coli* cells (Invitrogen, #C601003) were transformed with pAIDA_TEV_*fap2*EC, a positive colony was inoculated into 5 ml LB medium with 34 µg/ml chloramphenicol and grown overnight at 37 °C. 1 ml aliquots were frozen at −80 °C in 25% sterile glycerol for cryo-preservation. A 100 ml preculture in LB medium with 34 µg/ml chloramphenicol was inoculated from the cryostock and grown at 37 °C and 180 rpm overnight. The *E. coli* cells were then pelleted at $3300 \times g$ for 10 min and the supernatant was removed. 4 L of M9 minimal media supplemented with 5 mg/ml casamino acids, 100 µg/ml tryptophan and 35 µg/ml chloramphenicol were inoculated to an $OD_{600}$ of ~0.2 and incubated at 37 °C and 180 rpm. When an OD600 of ~0.7 was reached, the temperature was lowered to 15 °C and expression was induced at an $OD_{600}$ of ~0.9 with 0.5 mM isopropyl-β-D-1-thiogalactopyranoside (IPTG) (Roth, #367-93-1) for subsequent overnight expression. Cells were harvested by centrifugation at $3300 \times g$ for 9 min and resuspended in TEV cleavage buffer (20 mM TrisHCl pH 7.5, 200 mM NaCl, 10% glycerol, 0.5 mM EDTA, 1 mM DTT, 1 mM PMSF). GST tagged TEV protease (GST-TEV) was added to a concentration of ~70 µg/ml and the cell suspension was incubated for 4 h at 6 °C on a rotary incubator. The released Fap2-ECD was then separated from the cells by centrifugation at $50,000 \times g$ for 45 min. 10 mM $MgCl_2$, 40 mM imidazole and 200 mM NaCl were added before loading on a self-packed 3 ml Ni Sepharose™ 6 Fast Flow (Avantor, #17-5318-02) column using ÄKTA Pure FPLC system (Cytiva) at 4 °C. The bound protein was washed with 150 ml wash buffer (20 mM TrisHCl pH 7.5, 300 mM NaCl, 10% glycerol) and eluted with 36 ml elution buffer (20 mM TrisHCl pH 7.5, 300 mM NaCl, 10% glycerol, 300 mM imidazole). Collected fractions were analyzed by SDS-PAGE and Fap2-containing samples were pooled. Glutathione agarose resin (Cytiva, #17-0756-01) (1 ml) was used to remove remaining GST-TEV by batch incubation. Purified protein was either dialyzed into EM grid preparation buffer (20 mM TrisHCl pH 7.5, 200 mM NaCl) for vitrification or frozen in liquid nitrogen after adding glycerol to a concentration of 40% for storage at −80 °C.

## Expression and purification of hTIGIT-ECD

200 ml Expi293F™ cells (ThermoFisher Scientific (TFS), #A14527) were grown in Expi293™ Expression Medium (Gibco, #A14351-1) in two separate 500 ml suspension cell culture flasks (100 ml each) at 37 °C, 90 rpm and 8% $CO_2$. Transfection in each flask was performed at cell numbers between 1.5 and $2.0 \times 10^6$ per mL and a minimum cell viability of 95%, using 1.5 µg plasmid per ml cell suspension. The plasmid solution was diluted with Opti-MEM (reduced serum free medium, Gibco, #31985-047) to a volume of 2.08 ml, and 225 µl polyethylenimine (PEI; Polysciences, #24765-1) were added, followed by vortexing for 5 s. After incubation at 25 °C for 20 min, the DNA solution was added dropwise to the cell suspension. Incubation was continued at 37 °C, 90 rpm and 8% $CO_2$.

After 3 days, the cells were harvested by centrifugation at $500 \times g$ for 15 min at 4 °C and the pellet was frozen at −80 °C. The supernatant was collected and centrifuged for 30 min at $50,000 \times g$. Pierce™ protease inhibitor tablets (TFS, #A32963) were added to the supernatant and then dialyzed against 4 L of buffer W (50 mM TrisHCl pH 8, 200 mM NaCl) overnight. The cell pellets were thawed on ice, resuspended in 10 ml buffer W (50 mM TrisHCl pH 8, 200 mM NaCl) and lysed by dounce homogenization after addition of a Pierce™ protease inhibitor tablet. The lysate was centrifuged at $100000 \times g$ for 1 h at 4 °C and the cleared supernatant was pooled with the dialyzed medium and applied to a self-casted Strep-Tactin® Sepharose® (IBA, #21201010) gravity flow column containing 2 ml resin. The bound protein was washed with 25 column volumes (CVs) buffer W (50 mM TrisHCl pH 8, 200 mM NaCl) and eluted with 5 CVs Buffer E (50 mM TrisHCl pH 8, 200 mM NaCl, 2.5 mM desthiobiotin (IBA, #2-1000-002)). Elution fractions were analyzed for GFP emission signal, pooled accordingly and concentrated using Amicon®Ultra-4 centrifugal filter unit (Merck

Millipore, #UFC803024) to a final volume of 1 ml. Size exclusion chromatography was then performed with buffer W (50 mM TrisHCl pH 8, 200 mM NaCl) using a Superdex 200 Increase 10/300 GL column (TH Geyer, #28990944) to separate free eGFP from hTIGIT-ECD. Fractions containing hTIGIT-ECD were pooled from two runs and directly frozen in liquid nitrogen after addition of 40% glycerol for storage at −80 °C.

We also produced non-glycosylated hTIGIT-ECD in *E. coli* using a construct that comprises hTIGIT residues 23–128 with an N-terminal His$_6$-tag in pET19. Expression and purification from inclusion bodies were carried out as described previously[53] with the difference that the His$_6$-tag was not removed.

### Cell binding assays of Fap2-expressing *E. coli*

For quantitative cell binding experiments with HT-29 cells (ATCC, #HTB-38), ibidi μ-slide 8 well glass bottom plates (TFS, #17226213) were coated by adding 0.1 mg/ml poly-L-lysine (Biomol, #E-PB180523.10) to each well for 1 h at 37 °C with subsequent washing with sterile H$_2$O and drying for 2 h at 25 °C. 120,000 HT-29 cells were seeded in 300 μl DMEM/F12 (PAN-Biotech, #P04-41150) + 10% fetal bovine serum (FBS; PAN-Biotech, #P30-3302) per well and incubated at 37 °C and 5% CO$_2$ for 48 h.

The plasmids pAIDA_TEV *fap2*EC or pAIDA_TEV *fap2*EC_GalGalNAc_mut were used to express Fap2 in the OM of *E. coli* as described above. *E. coli* cells were harvested by centrifugation at 3200 × *g* for 5 min, washed three times with PBS pH 7.4 and adjusted to an OD$_{600}$ of 5.0 in 700 μl. Plasma membranes were stained by adding 3.5 μl CellBrite® Fix 555 (Biotium, #30088) for 30 min at 25 °C in the dark. The cells were harvested by centrifugation at 3200 × *g* for 5 min, washed once with PBS and resuspended to an OD$_{600}$ of 4.0. For assays in the presence of GalNAc, stained *E. coli* (OD$_{600}$ of 2.0) were incubated with 300 mM GalNAc (Glycon, #1811-31-0) for 30 min at 25 °C in the dark and diluted 1:1 with PBS upon adding 80 μl of the suspension to HT-29 cells. If no sugar was used, stained *E. coli* were adjusted to an OD$_{600}$ of 1 and directly added to HT-29 cells. As a control, *E. coli* cells were transformed using the empty pAIDA plasmid and treated similarly to the Fap2-expressing *E. coli*. After addition of *E. coli* to the HT-29 cells, they were incubated at 37 °C and 5% CO$_2$ for 1 h. The cells were then washed twice with PBS, fixed with paraformaldehyde (Roth, #0335.1) for 15 min at 25 °C and washed with PBS containing 1% Tween-20 (Roth, #9005-64-5) before staining with 150 μl Phalloidin-iFluor 488 (Abcam, #ab176753) for 20 min at 25 °C. After washing twice with PBS, the cells were imaged.

For binding analysis of Fap2 hTIGIT mutant-expressing *E. coli* to HEK293T cells (Merck, #12022001) transfected with pcDNA3.4_hTIGIT(1-162)_eGFP, 1.2 × 10$^6$ cells per well were cultured in 3 ml DMEM/F12 medium supplemented with 10% (v/v) FBS in 6-well plates (Sarstedt, #833920005). After 2 days the medium was replaced with 3 ml DMEM/F12 medium and the cells were transfected with 2 μg of pcDNA3.4_hTIGIT(1-162)_eGFP. The plasmid was diluted in 44.4 μl of pre-warmed Opti-MEM, PEI was added at a ratio of 2:1 (PEI:DNA), the solution was incubated at 25 °C for 20 min, and applied dropwise to the cell suspension. Cells were then cultured at 37 °C and 5% CO$_2$ and the medium was replaced after 24 h with fresh DMEM/F12 medium containing 10% (v/v) FBS and 3.75 mM valproic acid (VPA; Merck, #P4543). After 3 days of incubation the cells were resuspended in fresh DMEM/F12 medium with 10% (v/v) FBS, transferred to glass-bottom μ-Slide 8 well plates pre-coated with 0.1 mg/ml poly-L-lysine and incubated for 2 more days at 37 °C and 5% CO$_2$. The pAIDA_TEV *fap2*EC_hTIGIT_mut plasmid was used to express Fap2 on the OM in *E. coli*, the bacteria were stained and binding experiments were carried out as described above for the Gal-GalNAc mutant, with the difference that cell boundary staining was performed by using 150 μl of 0.1x CellMask™ Deep red (Abcam, #C10046) for 20 min at 25 °C.

The cells were imaged using a LSM780 confocal microscope equipped with spectral detector and PMTs (Carl Zeiss Microscopy). The LSM780 was controlled by the Zen Black software (Carl Zeiss Microscopy). To determine protein and *E. coli* localization, multicolor confocal imaging was performed in sequential mode with the following fluorophore-specific excitation (Ex.) and emission filter (EmF.) settings: eGFP (Ex.: 488 nm; EmF.: 490 - 520 nm), Phalloidin-iFluor 488 (Ex.: 488 nm; EmF.: 499–626 nm) and CellBriteFix® 555 (Ex.: 561 nm; EmF.: 545–649 nm). A Plan-Apochromat 40x/1.3 Oil DIC M27 objective (Carl Zeiss Microscopy) was used to acquire tile scans of 512 × 512 pixel images composed of z-stacks. Image analysis was performed using Fiji/ImageJ[81]. The collected tile scans were de-interleaved and stitched into z-stacks of 512 × 512 pixels containing the channels for eGFP/Phalloidin-iFluor 488 and CellBriteFix® 555 using a custom Fiji script. A Fiji macro, kindly provided by Dr. Tolga Soykan (Cellular Imaging Facility, Leibniz FMP), was modified and used to analyze the stitched images: The integrated fluorescence of stained *E. coli* was detected using the abundance of either Phalloidin for cross-correlation with cell boundaries or eGFP signal for cross-correlation with hTIGIT-eGFP expression as a prerequisite. Fluorescence measurements were then averaged per micrograph and statistically evaluated using the Games-Howell post-hoc test from the ggstatsplot package[82] in R.

Qualitative attachment assays of Fap2-expressing *E. coli* to Caco-2 (ATCC, #HTB-37) cells were conducted in a similar way as described for HT-29 cells with following deviations: The assay was performed in 24-well cell culture plates (Sarstedt, #833922005) with a total volume of 500 μl per well. Caco-2 cells were not stained and imaged in bright field.

### Labeling of Gal-GalNAc-Thr with AF 488-NHS and binding to Fap2-expressing *E. coli*

Gal-GalNAc-Thr (4.13 μmol; TCI #G0340) was dissolved in 200 mM bicarbonate buffer (1 ml, pH 8.3) and labeled with AF 488-NHS (Lumiprobe, #21820) dissolved in 10 μl DMSO in a 1:1.5 ratio for 1 h at 25 °C. Excess AF 488-NHS was removed after labeling by high-performance liquid chromatography (HPLC) using a preparative column (Reprospher 100 C18, 10 μm: 50 × 30 mm at 20 mL/min flow rate, linear gradient from H$_2$O/MeCN + 0.1 trifluoroacetic acid from 10 to 90% over 60 min) on a 1260 Infinity II LC System (Agilent). Peaks were collected manually according to the peak height at 480 nm. Separation was confirmed by mass detection in collected fractions using a single quadrupole LC/MSD system (Agilent) and fractions containing Gal-GalNAc-Thr-AF488 (0.45 μmol, yield: 10.9%, calc. for C39H43N4O23S2⁻ [M-H]: 999.18, found: 999.2, negative mode) were pooled and lyophilized for storage at −20 °C.

Fap2-expressing *E. coli* were prepared as described above. *E. coli* transformed with an empty pAIDA plasmid were used as a control. Gal-GalNAc-Thr-AF488 was dissolved in PBS pH 7.4 (200 μM) and added to the bacteria. The final concentration range was from 0 to 100 μM in a total volume of 400 μl with an *E. coli* density of OD$_{600}$ of 1.5. Bacteria were incubated for 1 h on ice, followed by 30 min at 37 °C and then pelleted for 7 min at 3,300 × *g*. After washing the pellet with PBS, it was resuspended in 160 μl lysis buffer (40 μl BugBuster (Merck Millipore, #71456) + 120 μl PBS). 0.5 μl Benzonase® (MoBiTec, #GE-NUC10700-01) was added and *E. coli* were lysed for 20 min at 37 °C, 600 rpm. The lysate was cleared by centrifugation at 16,000 × *g* for 20 min at 25 °C and fluorescence signal at 519 nm (excitation wavelength: 459 nm) was detected in a 96-well flat bottom microplate (Corning, #3686) with a Safire microplate reader (Tecan). Individual and mean fluorescence values of control and induced Fap2-expressing *E. coli* were visualized using the ggplot2 package in R[83].

### Pulldown assay of Fap2-ECD and hTIGIT-ECD

Purified Fap2-ECD (WT and hTIGIT binding site mutant) and hTIGIT-ECD-eGFP were thawed on ice from frozen aliquots (500 μl each). Fap2-

ECD was immobilized on 5% Ni-NTA magnetic beads (Cytiva, #28967390) for 2.5 h at 8 °C in a rotary shaker and washed three times with wash buffer (20 mM TrisHCl pH 7.5, 150 mM NaCl, 5 mM Imidazole). Thawed hTIGIT-ECD-eGFP was added to Fap2 immobilized on the magnetic beads and incubated for 2.5 h at 25 °C in a rotary shaker. Unbound hTIGIT-ECD-eGFP was removed by washing three times with wash buffer and the complex was then eluted from the beads with elution buffer (20 mM TrisHCl pH 7.5, 150 mM NaCl, 500 mM Imidazole). Pulldown assays using hTIGIT-ECD-Fc (Acro Biosystems, #TIT-H5254) were performed with a 2-fold molar excess of hTIGIT-ECD-Fc in the presence of 1x PBS pH 7.4 with 5 mM imidazole, followed by five wash steps (1x PBS, pH 7.4, 5 mM imidazole) after incubation for 2 h at 25 °C on a rotary shaker. For pulldown assays in the presence of Gal-NAc, 25 mM GalNAc was added to the incubation of Fap2 with hTIGIT and included in the wash buffer.

## Surface plasmon resonance of Fap2-ECD and hTIGIT-ECD

Binding kinetics were determined using a Biacore™ 3000 surface plasmon resonance system (Cytiva). hTIGIT-ECD in fusion with a C-terminal Fc tag (Acro Biosystems, #TIT-H5254) was immobilized on CM5 sensor chips (GE Healthcare) in 10 mM sodium acetate, pH 4.8, using a mixture of 0.2 M 1-ethyl-3-(3-dimethylaminopropyl)-carbo-diimide (EDC) and 50 mM N-hydroxysuccinimide (NHS) (Cytiva, #BR100050). The injection of 10 µg/ml hTIGIT-Fc for 7 min with a flow of 10 µl/min resulted in a final response unit (RU) value of 3220. Excess reactive groups were deactivated using 1 M ethanolamine-HCL, pH 8.5 (Cytiva, #BR100050). 50 µl Fap2-ECD was injected in different concentrations ranging from 52.5 to 368 nM and the response difference was recorded. All measurements were carried out in 25 mM HEPES-NaOH pH 7.5, 50 mM NaCl, and 0.005% Tween-20. Analyses were performed at 25 °C with a flow rate of 20 µl/min for 2.5 min and dissociation for 3 min. After each run, surfaces were regenerated with 30 µl 50 mM NaOH, 1 M NaCl at a flow rate of 20 µl/min for 1.5 min. The BIAevaluation software v 4.1.1 (Pharmacia Biosensor) was used to calculate association and dissociation rate constants by 1:1 Langmuir fitting of the primary sensorgram data.

## AlphaFold2 prediction of Fap2 ATCC23726, Fap2 ATCC25586 and RadD ATCC23726

A local installation of AlphaFold[45] was used to predict the structures of Fap2 (FusoPortal Gene 2068) and RadD (FusoPortal Gene 32) from *Fusobacterium nucleatum subsp. nucleatum* ATCC23726 and Fap2 from *Fusobacterium nucleatum subsp. nucleatum* ATCC25586 (FusoPortal Gene 1976). The search database status was Nov 17th, 2022. Due to the size of these proteins, predictions were carried out as separate parts with overlapping regions of at least 100 amino acids in the β-helices, and these were then covalently assembled in ChimeraX and energy minimized using ISOLDE[48]. For Fap2 (ATCC23726), fragments corresponding to residues 1–134, 90–245 + 2532–3271 including a GSG separator between the two parts, 242–2698, and 2800–3786 were predicted. For Fap2 (ATCC25586), fragments corresponding to residues 1–723, 300–928, 915–1638, 1625–2263, 2250–2962, 2949–3738 were predicted. For RadD (ATCC23726), fragments corresponding to residues 1–1164, 193–277 + 954–2388, and 98–199 + 2262–3461 were predicted.

## Negative stain EM

4 µl of either purified, detergent solubilized Fap2 from Fn ATCC25586 or recombinantly purified Fap2-ECD at a concentration of 0.05 mg/ml was applied to a freshly glow-discharged 300-mesh carbon-coated copper grid (Plano, #G2300C). After an incubation of 1 min, the sample was blotted with Whatman filter paper (Sigma, #WHA1004090), washed twice by application of 4 µl Milli-Q water with subsequent blotting and stained with 4 µl 2% uranyl acetate (Polysciences, #21447) solution for 1 min. The staining solution was removed with Whatman filter paper and grids were air-dried for at least 5 min at room temperature. Grids were imaged on a Talos L120 TEM (TFS) using a Ceta-16M CCD detector at 73,000x magnification.

## Cryo-EM sample preparation and data collection of Fap2-ECD and Fap2-ECD/hTIGIT-ECD

Purified Fap2-ECD at a concentration of 0.1 mg/ml was vitrified on Quantifoil R 2/1 Cu 300 mesh grids using a Vitrobot Mark IV (TFS) by applying 4 µl of sample twice with subsequent blotting for 2.0 s and 4.0 s, respectively, with a blotting force of -1, and 100% humidity at 12 °C.

Purified Fap2-ECD in complex with hTIGIT was eluted from magnetic beads and used for vitrification on Quantifoil R 2/1 Cu 300 mesh grids with a Vitrobot Mark IV (TFS). 4 µl of sample was applied 3 times with subsequent manual blotting steps using Whatman paper from the backside of the grid. Grids were vitrified after pipetting 4 µl buffer (20 mM Tris, 150 mM NaCl) onto the grid to reduce imidazole concentration. Blotting was done for 3.0 s, with a blot force of -1 and a humidity of 100% at 12 °C.

Micrographs were acquired on a Titan Krios G3i microscope (TFS) operated at 300 kV equipped with a K3 direct electron detector and Bioquantum energy filter (Gatan, Digital Micrograph version 3.32.2403.0) running in CDS superresolution counting mode at a slit width of 20 eV and at a nominal magnification of 81,000×, giving a calibrated pixel size of 0.53 Å/px. For Fap2-ECD, movies were recorded for 2.0 s accumulating a total electron dose of 59.8 e⁻/Å² fractionated into 53 frames. For Fap2-ECD/hTIGIT-ECD, movies were recorded for 2.0 s accumulating a total electron dose of 44.6 e⁻/Å² fractionated into 38 frames. EPU (Version 3.3) was utilized for automated data acquisition using nominal defocus values between −1.5 and −2.6 µm for Fap2-ECD, and between −1.0 and −2.5 µm for Fap2/hTIGIT-ECD, respectively.

## Cryo-EM data analysis of Fap2-ECD

Processing of the cryo-EM data was performed in CryoSPARC[84] and is outlined in Supplementary Fig. 4a. 12,663 movies were collected, aligned using Patch Motion correction and CTF was determined with Patch CTF estimation. After sorting out bad images, 12,391 micrographs were kept and processed further. A small subset of 59 micrographs was used to manually pick 474 particles to train Topaz[85]. 330,980 particles were then picked using the trained Topaz model, extracted (7x binned), and subjected to 2D classification (30 online-EM iterations). This process was iterated by repeating twice, using 30,000 and 52,444 particles of selected 2D classes to train new Topaz models. The final round of 2D classification yielded 321,160 particles after extraction from micrographs without binning (1.06 Å/px) which were used for an ab-initio reconstruction. Subsequent homogeneous and non-uniform refinement followed, the latter using the automatically generated static mask from the homogeneous refinement. Particles were then subjected to two further rounds of 2D classification (30 online-EM iterations each), leaving 265,956 particles. These were used for non-uniform refinement of the previously obtained volume, which resulted in a map with a resolution of 4.7 Å according to the gold-standard Fourier shell correlation (FSC) criterion (Supplementary Fig. 3b, Supplementary Table 1).

To optimize resolution in the N-terminal matchstick region, the following steps were carried out: The volume from the first non-uniform refinement was used to fit the AlphaFold2 prediction of Fap2 from Fn ATCC23726 (residues 42–2698) into the density. The model was automatically fitted using the flexible molecular dynamics fitting software Namdinator[47] and manually adjusted in Coot[86]. The Fap2 model was then assembled in ChimeraX[87] based on an AlphaFold2 prediction to include a C-terminal section of the Fap2-ECD construct that could not be fitted into the density. The resulting model of Fap2-ECD (42–3271) was then used to generate an artificial map in ChimeraX with a resolution of 5 Å, which was applied as a template for a non-

uniform refinement, using the cleaned-up particles from the final 2D classification. A mask (Dilation radius 8, soft padding width 15) covering the N-terminal end was generated in ChimeraX, followed by 3D classification to find particles with good alignment on the matchstick region. 160,806 particles were retained and subjected to particle subtraction of the C-terminal region using a mask (Dilation radius 10, soft padding width 20) generated in ChimeraX. 160,676 subtracted particles were used for local refinement, and the resulting volume and particles were re-aligned with the volume alignment tools job to shift the N-terminal matchstick region to the center of the box (new center: 350, 350, 417; box of 700 px). A local refinement job was carried out using the shifted particles, and particles connected to the resulting volume were re-extracted from micrographs with a box size of 384 px. A mask (Dilation radius 6, soft padding width 12) covering most of the N-terminal region was generated in ChimeraX, followed by local refinement and 3D classification using the subtracted particles. 151,825 particles were selected and a final local refinement resulted in a map with a global resolution of 4.4 Å, according to the gold-standard FSC criterion (Fig. 1g, Supplementary Fig. 3e). DeepEMhancer[88] was applied for map sharpening. In addition, particles underlying the last refinement were imported in Relion[89] using pyEM[90], refined and post-processed, resulting in a map with 4.4 Å resolution according to the gold-standard FSC criterion.

### Modeling of membrane distal part of Fap2-ECD
Fap2-ECD (42-2698) that has been fitted to the full-length Fap2-ECD map using Namdinator (see above) was trimmed to residues 283–1817 and rigid-body fitted in the 4.4 Å density map using ChimeraX, followed by one further MD simulation based fit in Namdinator. The resulting model was manually adjusted in ISOLDE[48]. The limited resolution allowed reasonable fit of the unstructured N-terminus that complements the hydrophobic longitudinal groove and the extensions from the β-helix at the membrane-distal tip (Fig. 1h), although the individual β-strands of the β-helix were not separated in the density map. The model was then further trimmed to residues 290–1772 in Coot[86], and iteratively refined using Phenix real-space refinement and ISOLDE. The final model parameters are shown in Supplementary Table 1.

### Cryo-EM data analysis of Fap2-ECD/hTIGIT-ECD
Processing of the cryo-EM data was performed in CryoSPARC and is outlined in Supplementary Fig. 8f. Two data sets were acquired and combined, resulting in 15,658 patch CTF-estimated, motion-corrected micrographs after discarding low-quality images. Initially, 455 particles were picked manually on 49 micrographs to train Topaz. The trained model was used to extract 659,696 particles from micrographs with 3x binning, followed by 2D classification, and 41,675 particles were selected to train Topaz again. Subsequent picking resulted in 1,191,741 particles, which were extracted from micrographs (4x binning) and applied to two rounds of 2D classification (50 online-EM iterations). An ab-initio reconstruction (5 classes) of the remaining 761,580 particles was used for an initial 3D sorting step, and 622,717 particles were retained for extraction (no binning) and two iterative rounds of 2D classification (50 online-EM iterations). 73,482 particles were selected for an ab-initio reconstruction, which was used for subsequent homogeneous and non-uniform refinement, the latter using the mask from the homogeneous refinement. The volume alignment tools job was applied to re-align particles to shift the N-terminal matchstick region to the center of the box (new center: 350, 350, 210; box of 700 px). The re-centered particles were used for an ab-initio reconstruction and further refined with subsequent homogeneous and non-uniform refinement, with the mask generated in homogeneous refinement. The resulting volume was used to create a mask in ChimeraX covering the matchstick region, with a dilation radius of 6 and soft padding width of 20. The N-terminal focus mask was applied for two independent 3D

classifications in simple and input modes, respectively, both with hard classification enabled. Classes were selected based on the presence of extra density at the N-terminus, yielding 45,207 particles. These were further processed using non-uniform refinement to a final global resolution of 6.0 Å according to the gold-standard FSC criterion (Supplementary Fig. 8c, Supplementary Table 1).

### Cultivation of Fn ATCC25586
*F. nucleatum* ATCC25586 (Leibniz Institute DSMZ, German Collection of Microorganisms and Cell Cultures) was grown at 37 °C under anaerobic conditions in reduced Schaedler broth (Sifin diagnostics #TN1203). After overnight culture and confirmation of bacterial growth by OD measurement, the bacteria were harvested at $3000 \times g$ for 10 min and the supernatant was replaced with fresh medium. 5 ml of Jurkat cells (Merck, #88042803; $3.2 \times 10^6$ cells/ml) were added to 45 ml of Fn culture and incubated overnight at 37 °C under anaerobic conditions to stimulate Fap2 expression of Fn, as described previously[40]. The cell suspension was harvested the next day at $3000 \times g$ for 10 min and frozen at −80 °C.

### Cryo-ET of Fn ATCC25586 and tomogram reconstruction
A frozen sample of *Fusobacterium nucleatum* ATCC25586 cells was thawed and diluted in 1x PBS pH 7.4 to an $OD_{600}$ of -0.3. 4 µl of the bacterial suspension was mixed with 10 nm gold fiducials, added to a R1.2/1.3 300 mesh grid (Quantifoil, #N1-C14nCu30-01) and blotted for 3.0 s, with a blot force of 3 and humidity of 100% at 12 °C using a Vitrobot Mark IV (TFS). Imaging was performed on a FEI Titan Krios G3i microscope (TFS) operated at 300 kV equipped with a K3 direct electron detector and Bioquantum energy filter (Gatan, Digital Micrograph version 3.32.2403.0) running in CDS counting mode at a slit width of 20 eV, operated by SerialEM v4.0.4[91]. A nominal magnification of 53,000x was used, which resulted in a calibrated pixel size of 1.68 Å/px. Tilt series were acquired using a dose-symmetric scheme[92], with a tilt axis angle of 84.4°, 3° angular increment and target defocus of −5.5 µm. Tilt series were recorded as movies of 10 frames in counting mode and a dose rate of 5.65 e$^-$/Å$^2$/sec resulting in a total dose of 160.7 e$^-$/Å$^2$ per tilt series. Frames were aligned using Warp/M[93]. Tomograms were reconstructed in IMOD (v.4.11.24)[94], using 'ctf phase flip' for CTF correction and back projection with Simultaneous Iterative Reconstruction Technique (SIRT)-like filter equivalent to 15 iterations.

### Isolation of Fn adhesins from outer membrane
Frozen suspensions of *F. nucleatum* sp. ATCC25586 were thawed, resuspended in lysis buffer (50 mM Tris-HCl pH 7.2, 300 mM NaCl, 10% glycerol), and lysed using a LM10 Microfluidizer (Microfluidics) at 15,000 psi and 4 °C, followed by removal of cell debris via centrifugation for 30 min at $50,000 \times g$ and 4 °C. The membranes in the cleared lysate were pelleted for 2 h at $100,000 \times g$ and 4 °C. Membrane pellets were homogenized in lysis buffer (50 mM Tris-HCl pH 7.2, 300 mM NaCl, 10% glycerol) using dounce homogenization, and 1% n-dodecyl β-D-maltoside (DDM; Glycon, #69227-93-6) was added for solubilization overnight at 8 °C while shaking. Non-solubilized material was removed by centrifugation at $180,000 \times g$ and 4 °C for 1 h, and the supernatant was loaded on a Superose 6 10/300 column, using a buffer containing 50 mM Tris-HCl pH 7.2, 300 mM NaCl, 10% glycerol and 0.05% DDM. Collected fractions were analyzed by SDS-PAGE and negative stain EM.

### Peptide fingerprint of purified Fap2-ECD, and Fap2 and RadD from Fn ATCC25586
Purified recombinant Fap2-ECD was further purified on a HiLoad 16/600 Superose 6 pg column (Cytiva, #29-3239-52) equilibrated in 50 mM Tris-HCl pH 7.4, 300 mM NaCl, 10% glycerol before MS analysis. Isolated OM proteins from Fn ATCC25586 were used after the Superose 6 (see above). Proteins were loaded on SDS-PAGE and subjected to

in-gel digestion. For this, gel bands were excised, reduced with 5 mM DTT at 56 °C for 30 min and alkylated with 40 mM chloroacetamide at room temperature for 30 min in the dark. Protein digestion was carried out using trypsin at an enzyme-to-protein ratio of 1:100 (w/w) at 37 °C overnight. LC-MS analysis was performed using an UltiMate 3000 RSLC nano LC system coupled on-line to an Orbitrap Fusion mass spectrometer (TFS). Reversed-phase separation was performed using a 50 cm analytical column (in-house packed with Poroshell 120 EC-C18, 2.7 μm, Agilent Technologies) with a 120 min gradient. MS1 scans were performed in the Orbitrap using 120,000 resolution; MS2 scans were acquired in the ion trap with an AGC target of 10,000 and maximum injection time of 35 ms, charge state 2–4 enable for MS2. Data analysis including label free quantification was performed using MaxQuant (version 1.6.2.6 and 2.0.3.0) using the following parameters: MS ion mass tolerance: 4.5 ppm; MS2 ion mass tolerance: 0.5 Da; variable modification: Met oxidation, Acetyl (Protein N-term), Cys carbamido-methyl; protease: Trypsin (R,K); allowed number of missed-cleavages: 2, database: SwissProt and *E. coli* database (SwissProt_14dez, *Escherichia_coli_2020*) combined with the sequences Fap2-ECD from Fn ATCC23726 and TEV protease, as well as Fap2 and RadD from Fn ATCC25586; label free quantification; match between runs disabled. Results were reported at 1% false discovery rate at the protein level.

### Analysis of post-translational modification (PTM) of Fap2 ATCC23726 expressed in *E. coli* and Fap2 ATCC25586 from *Fusobacterium nucleatum*

Proteins were cut from SDS-PAGE and digested into peptides in solution using 8 M Urea in 50 mM TEAB pH 8.5, supplemented with 5 mM TCEP and 40 mM CAA, 1:200 (enzyme:protein ratio) LysC and 1:100 Trypsin for 16 h at 37 °C. Peptides were acidified to 1% formic acid and desalted with C18 StageTip. Desalted peptides were resuspended in 1% acetonitrile (ACN) with 0.05% trifluoracetic acid and 1 μg was injected into a Dionex™ UltiMate™ 3000 system (TFS) connected to a PepMap C-18 trap-column (0.075 mm × 50 mm, 3 μm particle size, 100 Å pore size, TFS) followed by an in-house packed C18 column for reverse phase separation (Poroshell 120 EC-C18, 2.7 μm, Agilent Technologies). Peptides were separated using a 117- or 177-min gradient with a flow rate of 250 nL/min with increasing ACN concentration and analyzed on an Orbitrap Fusion mass spectrometer (TFS) and Instrument Control Software version 4.0. MS1 scans were acquired in the Orbitrap with a mass resolution of 120,000. MS1 parameters were as follows: scan range m/z 375–1500, standard AGC target, 50 ms maximum injection time. MS2 scans were acquired in the Orbitrap with the following parameters: 15,000 mass resolution, first mass 110 m/z, standard AGC target, 22 ms maximum injection time, isolation window 1.6 m/z, NCE 30%. Previously isolated precursors were excluded from fragmentation for 40 s. Only precursors with charges +2 to +4 were subjected to MS2. The raw data files were analyzed using the default open-search workflow available in FragPipe version 22.0[95,96]. The analysis parameters included a precursor mass offset range of −150 to +1000 Da, an MS2 tolerance of 20 ppm, a minimum peptide length of 5 and a minimum peptide mass of 500 Da. Variable modifications considered were oxidation of methionine and acetylation at the protein N-terminus, while carbamidomethylation of cysteine was included as a fixed modification. The data were searched against the sequence of Fap2 along with a database of common contaminants. Unknown modifications were identified from the global.modsummary file and data were visualized using the ggplot2 package in R[83].

### Molecular docking of hTIGIT-ECD to Fap2-ECD

The Fap2-ECD model (residues 242–2698) was rigid-body fitted in the cryo-EM density map of the Fap2-ECD/hTIGIT-ECD co-structure using UCSF ChimeraX. Molecular docking of hTIGIT-ECD (residues 22–141, PDB ID 3UCR) was carried out with the HADDOCK2.2 (High Ambiguity Driven protein-protein DOCKing) web server[59]. Amino acids forming the distal surface of the matchstick region (Fap2 residues 362–373, 533–544, 548–551, 594–595, and 720–721) were selected for the docking in accordance to the extra density observed for the Fap2-hTIGIT co-structure in comparison to the structure of Fap2 alone (Fig. 4b, c). The cluster with the best docking results (Supplementary Table 4) was further analyzed in MD simulation.

### Molecular docking of Gal-GalNAc-Thr to Fap2-ECD

Gal-GalNAc-Thr was docked to the AlphaFold2 predicted Fap2 model (339–910) using Autodock Vina[97] in UCSF Chimera (V1.17.1)[98]. Protonation of Fap2 was set to charge Asp, Glu, and Lys. The search frame was set to include the matchstick region (residues 339–910) at the distal end of Fap2. The best docking pose gave a free energy score of −6.1 kcal/mol (Supplementary Fig. 10a). To confirm the relevance of the docking, it was performed in three replicates searching for ten states each on a larger N-terminal area of Fap2 (312–1381) and compared to docking results obtained for the N-terminal region of RadD (474–1274). The docking results were visualized in ChimeraX and colored according to the docking score (Supplementary Fig. 10b,c).

### MD simulation

The best molecular docking results were analyzed by MD simulations using the GROMACS 2022 package[99] with the CHARMM36 force field, which is compatible with proteins and glycoproteins[100], and the standard water model CHARMM-modified TIP3P[101].

For the simulation of the Fap2/Gal-GalNAc-Thr complex, the docked structure was split into receptor and ligand structure for further processing. The receptor topology was generated using the *pdb2gmx* program with interactive selection of NH3+ and COO- termini. This manual selection of protein-specific termini is necessary due to the N-terminal methionine residue for which *pdb2gmx* selects the incompatible residue type t by default. The docked ligand structure was hydrogenated and converted to (.mol2) format using UCSF Chimera[98]. The ligand topology was generated using the online tool SwissParam[102]. The [atomtypes] and [pairtypes] entries from the resulting.itp file were transferred to the resulting (.prm) ligand file, which was then converted to the GROMACS (.gro) format using *gmx editconf*. Finally, the receptor and ligand.gro files were merged and the ligand was added to the topology file. For the simulation of the Fap2-hTIGIT complex, the topology was generated directly from the complete docked structure. Supplementary Table 8 shows an overview of the system setup.

The complex was immersed in a dodecahedral or cubic box with edge spacing of 1 nm. The box was filled with spc216 water molecules and sodium ions to neutral pH using *gmx editconf*, *solvate*, *grompp* and *genion*. Subsequent energy minimization, NVT and NPT equilibration, and production simulation were performed with 10 replicates, yielding 10 separate systems. The systems were energy minimized using the steepest integrator and Verlet cutoff scheme. NVT equilibration was performed using the md leap-frog integrator in 50,000 2-fs steps with V-scale temperature coupling between protein and non-protein groups with a 300 K reference temperature. NPT equilibrations were performed using Parrinello-Rahman pressure coupling. A Verlet cutoff scheme was used for van der Waals interactions and the Ewald particle mesh for long-range electrostatic interactions. No distance constraints were applied. The production simulations were performed with the same parameters for 150 ns. Initial velocity generation was left disabled as the separate energy minimization, NVT and NPT equilibration already introduced variation in the initial system conformation between individual replicates.

The resulting trajectories were fixed and centered with *gmx trjconv* and analyzed with the VMD hbonds plugin 1.2[103] and the GROMACS-implemented RMSD tool[99] to detect movements of the ligand relative to the binding interface. For the latter, the C-alpha backbones of the ligand and receptor, respectively, were grouped in

the index and subsequently selected for RMSD analysis. RMSD plots are depicted in Supplementary Fig. 11. Hydrogen bond analysis was performed by selecting all receptor residues within 5 Å of the ligand in the selected docking state as selection 1 and the ligand as selection 2. A distance cutoff of 3.5 Å and an angle cutoff of 35° were used, and the results of multiple replicate trajectories were statistically analyzed using a custom Python script. The most consistent interaction partners were visually inspected in VMD[103]. Representative trajectories were rendered using the VMD Movie Maker plugin[103] combined with FFmpeg[104].

### Binding site prediction of Fap2-ECD matchstick region
The neural network based binding site predictor DeepSite[61] was used to predict binding sites of the Fap2 matchstick region. An AlphaFold2 prediction of Fap2 (339–910) was used for analysis on the playmolecule web-based server (www.playmolecule.org, accessed on Jan 17th, 2024) and results were visualized with ChimeraX, coloring the predicted sites according to their score.

### Reporting summary
Further information on research design is available in the Nature Portfolio Reporting Summary linked to this article.

## Data availability
Cryo-EM maps have been deposited in the Electron Microscopy Data Bank (EMDB) under accession numbers EMD-53048 (Fap2-ECD), EMD-53049 (membrane-distal part of Fap2-ECD), and EMD-53052 (Fap2-ECD/hTIGIT-ECD complex), respectively. The atomic coordinates of Fap2-ECD (290-1772) have been deposited in the PDB under accession code 9QE7. The atomic coordinates of the TIGIT IgV domain with PDB accession code 3UCR have been used for modeling. The source data underlying Figs. 1a, e, 2a, c, 3a, c, d, 4f, 5d, e and Supplementary Figs. 2b, c, g, h, 4b, 5a, b, d, f, 7a, c–g, 9a, and 11a-d are provided as a Source Data file. The cryo-EM data sets generated in this work are available from the corresponding author on request. Source data are provided with this paper.

## Code availability
The MD simulation script files are available within the Source Data files. The scripts for automated statistical evaluation of fluorescence micrographs and visualization of the plots are provided with the Source Data files.

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

## Acknowledgements

We thank Uwe Fink, Alina Roderer (both Leibniz FMP), and Julia Schmidt (Charité—Universitätsmedizin Berlin) for excellent technical support, and Marianne Schoenfelder (TU Berlin) for assistance in TIGIT purification. We thank Dr. Oxana Krylova (Biophysics facility, Leibniz FMP) for support with recording and analyzing SPR data. We thank Dr. Johannes Broichhagen, Dr. Blaise Gatin-Fraudet, and Kilian Rossmann (ChemBioProbes, Leibniz FMP) for their help with Gal-GalNAc-Thr labeling and purification, and Dr. Peter Schmieder and Nils Trieloff (NMR Core facility, Leibniz FMP) for NMR analysis of labeled Gal-GalNAc-Thr. We acknowledge peptide fingerprint analysis carried out and analyzed by Heike Stephanowitz and the PTM analysis performed by Dr. Max Ruwolt (Mass Spectrometry Core facility, Leibniz FMP). We acknowledge training for and access to the Zeiss confocal microscope by Marie Bieck, Dr. Tolga Soykan, and Dr. Martin Lehmann (Cellular imaging facility, Leibniz FMP). We thank Dr. Tillmann Utesch (Structural Chemistry and Computational Biophysics, Leibniz FMP) for help in setting up the MD simulations. We acknowledge access to electron microscopic equipment at the Core Facility for cryo-Electron Microscopy (CFcryoEM) of the Charité—Universitätsmedizin Berlin supported by DFG (INST 335/588-1 FUGG) for cryo-EM and cryo-ET data collection, and we thank Dr. Christoph Diebolder (CFCryoEM) for support in cryo-ET data analysis. Barth van Rossum (Leibniz FMP) is acknowledged for the artwork in Fig. 6. This work received funding from the Deutsche Forschungsgemeinschaft (DFG, project 548509121, to D.R.).

## Author contributions

F.S. designed expression constructs, carried out and evaluated Fap2 production, cryo-EM, cryo-ET, and functional assays. G.L.M. carried out and analyzed MD simulations. F.S. and K.M. carried out and analyzed cell adhesion assays. T.S. recorded cryo-EM and cryo-ET data, including data preprocessing. A.M. supervised and J.K. carried out and optimized Fn cultivation. D.R. designed and supervised research and carried out docking and modeling, together with F.S. D.R., F.S., and G.L.M. prepared figures, movies, and wrote the manuscript with contributions from all authors.

## Funding

## Competing interests

J.K. is CEO of MoKi Analytics GmbH. A.M. and J.K. are shareholders of MoKi Analytics GmbH. A.M. is the owner of the private practice Moter Diagnostics. The other authors declare no competing interests.
