## [Transparent Peer Review file · Nature Communications]

Structural basis of *Fusobacterium nucleatum* adhesin Fap2 interaction with receptors on cancer and immune cells

Corresponding Author: Dr Daniel Roderer

Version 0:

Reviewer comments:

Reviewer #1

(Remarks to the Author)

Review Structural basis of *Fusobacterium nucleatum* adhesin Fap2 interaction with receptors on 2 cancer and immune cells
In the current study Roderer and colleagues provided the structure function analysis of the adhesin Fap2 from the Gram negative *Fusobacterium nucleatum*. The purified the adhesin after heterologous expression in *E. Coli*, tested its interaction with CRC receptor glycan Gal-GalNAc and T-cell marker hTIGIT. The authors used cryoEM in combination with fluorescence microscopy and other spectroscopic methods. Based on their results they proposed a model for the role of Fap2 in bacteria binding to tumors and deactivation of responding NK cells
Overall the manuscript reads quite interesting and the results deserve publication in Nature Communications. There is however some major and minor issues that need to be solved before acceptance of the manuscript.

Major concerns

My first major comment is regarding the data presented in Fig3. Experiments shown in this Fig show necessary results. For their argumentation it is necessary to show the co-localization of the ligand with the Fap2 using two different fluorescent markers

My other concern is with regard to the interesting results presented in Fig4. Those structural and modeling results need to be verified by mutational analysis of residues that were found to be specifically involved in the intermolecular interaction
Finally, I was wondering about the obtained resolution for the EM maps presented in the current study. I think the presented data is still structural heterogeneous. The authors should comment if they have fine-tuned further "data polishing" by removing noise, and extended 2D classification or if they have considered other strategies.

Minor concerns

Line 78 to 91: Why too detailed? I appreciate that the authors recap the current situation and define the open questions in the field. The many details (starting from line 83 until the end of the introduction) however, should be presented in the discussion and not in the introducing paragraphs.

Line 94: title does not reflect the content of this chapter

Line 114-16: authors are not precise here. Moreover, showing an alignment of two orthologous genes to argue about conservation aspects is not convincing. Authors might add more sequences to the alignment or they delete this comment.
fig S1: How does the alignment in panel B relate to the structures shown in A? Labelling of the region including residues from 300 to 1000 would be great

Line 126: and Fig1 do not match. Apparent molecular mass can differ sometimes a bit but the observed difference of 50 to 60kDa seems very high. The authors should verify the length of the purified protein by tryptic digestions and mass spectrometry or another method that reveals sequence details

Line 140: the authors do not give any details on the quality of the obtained fitting results. Add please.

Line 140-43: I cannot follow the authors argumentation checking the panels I and J of Fig1. The authors should explain why in panel I two different model fittings are presented to the reader and how does this relate to panel J. Taken together, panel G to J are a bit redundant and the point with the hydrophobic groove is poorly illustrated. Some of the panels could go into

supplement.

Line 149-50: This comparison is invalid. Fig1 J is showing an EM map while SupplFig1 shows an alphaFold model

Line 152: change the text to
putative receptor-binding β -helical ECD

Line 169-70: The given argumentation is not convincing in light of Fig2 panel C. Indeed, some structures match while other do not. Did the authors tried comparison of the observed cryoET species with SPA 2D class averages?

Line 170-72: The negatively stained specimen shown in SIFig. 5e are not conclusive.

Line 187-89: I do not understand what is shown here. The authors might revise the plot shown in panel A. What is the control in this experiments, how were the data fitted, how was the affinity calculated? Explain and show data, please

Line 191-94: The argumentation and the corresponding data presented in Fig3 look nice. However, the authors should consider co-localization studies using an additional fluorescent marker for Fap2 or they have to tone down their argumentation a bit.

Line 216-24: I have problems understanding the co-purification illustrated in SFig7. In corresponding panels a to c the hTIGIT construct has a molecular mass of about 40kDa while in the pull-down experiments shown in panel d the mass is almost 20% higher. The authors should explain this mass difference

Line 229-31: SPR spectra seem to be sub-optimal (baseline shift, bulk shift which could have several reasons. Moreover the authors do not show here or elsewhere in the supplement the R_u as a function of concentration. This data is essential to verify the quality of the SPR experiments

Line 243-50: Check my comments on Fig. 4 (see below)

I see two questions that need to be addressed by the authors.

Is there an additional density at the tip/

is there further structural differences when hTIGIT binds?

The later questions was not at all addressed. The authors should comment on this issue.

Panel a to d of Fig4 is pretty redundant. One panel in main fig. Might be enough. Rest of the data can go to the supplement.

Fig1: color code is confusing: panel B and D use similar color for different regions

Too many different font (& size) used

panel B: linker is not part of the passenger domain

panel I and J: what is the point with the hydrophobicity/hydrophilicity? This figure is not helpful to follow the authors argumentation

Fig2. Panel B: OM looks corrupted. The authors might comment on this

Fig5: Not conclusive: Does that imply that FN simply forms a distance barrier?

FigS1: label N- and C-terminus, correlate panels A and B

Reviewer #2

(Remarks to the Author)

The manuscript «Structural basis of *Fusobacterium nucleatum* adhesin Fap2 interaction with receptors on cancer and immune cells» by Schoepf et al. describes structural and functional details of Fap2 binding to its two main receptors, one a signaling receptor of cancer cells, and the other a carbohydrate component of the extracellular matrix. The manuscript is generally well-written, and I have only a few major comments (and minor comments mostly relating to language)

Major comments:

1) In different places in the text, the authors suggest directly or indirectly that they are the first to purify a *Fusobacterial* autotransporter, and/or neglect to mention that parts of their methodology has been used successfully previously in related experimental systems. I am not suggesting that this is deliberate (it's oversights – it is a complex field, and purification strategies are not always obvious from abstracts); nevertheless, I suggest to modify some claims made related to the protein purification strategy, and to amend some text with additional citations:

- Abstract p1 line 33: “first purification”. A Google Scholar search with the terms “purification of fusobacterial autotransporter” immediately gives a long list of fusobacterial autotransporters that have been purified for different purposes before. I suggest to simply delete the complete last sentence of the abstract.

- Introduction p3 line 88-91. AIDA-I based expression systems have been around for a long time, also for expressing heterologous proteins on the E.coli surface (people used to call this “autodisplay” in literature. Check “AIDA-I heterologous

expression" in Google Scholar). Likewise, proteolytic cleavage of passenger domains of autotransporters is not an original invention of the authors. A significant number of bacterial autotransporters are autocatalytic proteases that release their passenger domain by default (might be worth mentioning). For an example of an autotransporter with an engineered protease cleavage site, see f.ex. www.nature.com/articles/srep28020 ... I suggest to rephrase "our system" to maybe "our strategy" and maybe to add more information on prior work.

- Results p4 line 117: here, the autodisplay system using AIDA-I is mentioned for the first time. The paragraph should reference to the original work where this was developed – not only to give credit to the developers, but also to explain better what is meant with "are designed to enable translocation of heterologous cargoes..."

- Somewhat related to this: P5 line 173-175: "...our findings suggest that a large number of Fap2 and RadD molecules are present in the Fn OM, where the linker domains allow flexible attachment". The surface density of these two molecules was most probably quantified by others previously, and if so, this needs citations (and then the findings "confirm", not "suggest"). As a separate point, it might be good to repeat more clearly here what is defined as the linker domain, and what experimental data the authors mean when they claim that their results are in line with "flexible attachment" (and I assume they mean attachment to host cells, this could be made more explicit too).

- Discussion P9 line 319-331 – this section would benefit from comparison with surface display of other autotransporters from different bacterial species (with citations... see above), rather than a collection of toxins. Also, I do not see the point of the size discussion here. I am not aware of any size limit for the autodisplay systems (using AIDA-I or any other autotransporter). Proteins of any size should potentially be "expressable/exportable" through these systems. Known limitations are charge, premature/fast folding, and potentially, proline residues or other sequence features that might hinder export through a narrow pore.

2) I find the binding of Fap2 to two different receptors interesting, but I do think that some of the data needs better controls (with minimal effort and using methodology the authors already employ, see below). In detail, what I mean is:

- P5 line 182-183 (and later P6 line 216): here the authors themselves suggest that Gal-GalNAc is part of complex glycosylation patterns (true). They later state that hTIGIT is glycosylated (see below). What I am really missing is both a discussion, and experimental controls that the binding that is observed to Gal-GalNAc and to hTIGIT is possibly the same thing. Or can the authors be very sure that glycosylated hTIGIT does not have a Gal-GalNAc motif in its glycans?

- Based on the above, I am really curious why the authors did not do the obvious control experiment: a competition experiment where they "interfere" with Fap2-hTIGIT binding in the presence of increasing amounts of Gal-GalNAc. This should be done both with pulldowns and SPR, and probably also in the cell-binding experiments, in the same way they did the original binding studies to hTIGIT (page 6/7). If the authors are correct, then there should be no competition. If there is, then their presumed binding sites are at least overlapping. In any case, this control experiment would confirm (or not) the trustworthiness of the MD/docking studies.

- This obviously then would relate to changes in the discussion, especially around page 10 line 353.

Minor comments in order of appearance:

Abstract

P1 line 19: "numerous microbiota". I am unsure that this is correct terminology. In my view, we are talking about dysbiotic situations here, and not about a subset of healthy interstitial microbiomes/microbiota (microbiota? Is that the correct plural?)

Introduction

P2 line 43: the third-most, the second-most... (needs hyphens). See also line 58 CRC-driving...

P2 line 63: "such as by" – poor grammar, please rephrase?

P3 line 76: how is it relevant here that there is no structure of "another" autotransporter from Fn available? Please explain or delete.

Results

P3 line 95-104. Nothing in this paragraph is results. This should be part of the introduction I think. I also do not see the value of Figure 1a to the results section of the manuscript – The genomic localization of different Fn autotransporters is not discussed anywhere later, and does not link to any result?

P4 line 127: how was ">90% purity" quantified? Or is this just an "eye-balled" thing based on an SDS gel (then rephrase)?

P4 line 134: "parted in two branches". I find this expression confusing. "branches" to me suggest a 'forking' of sorts (which is not the case here). What the authors are trying to express is that there is a kink in an otherwise pretty linear, rod-like structure. Certainly the 'kink' is not the origin of two branches. If this is expression changed here (and I think it should be), make sure that this is also checked in other parts of the manuscript.

P4 line 142: "this gap is closed..." – what gap? Do you mean the "groove" from the sentence before?

P4 line 143-145: "The unstructured N-terminus is highly conserved in different fusobacterial species, highlighting also the conservation of the hydrophobic groove,..." – if it has a groove, how can it be unstructured? I am under the impression that this needs better explanation.

P5 line 151 "our findings report..." findings cannot report.

P5 line 152-154: the authors "conclude" here that their autotransporter acts as a rod-like adhesin "in functional analogy" to other similarly shaped autotransporters. I think this is a bit mis-phrased. First of all, this is probably homology, not analogy. Secondly, it is not really clear to me where the "conclusion" comes from? From comparing shape?

P5 line 164: "both of them being the largest ORFs in Fn (Fig 1a)". This should probably come with a citation as it is not an analysis performed by the authors. Also, Fig 1a has nothing to do with ORF length?

P5 line 171: "the cell attachments keep their shape and identity as indicated by SDS PAGE" – I am not convinced that SDS page can show that? Consider rephrasing.

P6 line 188: "pAIDA without Fap2" is presumably pAIDA wildtype (i.e. expresses AIDA-I)? Same line: "at receptor concentrations as low as 23-100 μM " - please be more explicit that this refers to the disaccharide (it does, I think).

P6 line 196/197: "This indicated that surface Gal-GalNAc is not equally accessible to the bacteria within the cell population" – compared to what? And can the authors be sure that surface Gal-GalNAc is evenly distributed on the cells and that this is the reason for uneven distribution of bacteria? I also miss a discussion of the possibility that the bacteria autoaggregate (this is quite common when they express autotransporter adhesins – even though I cannot say for Fap2) and that this is the reason for the observed clustering.

P6 line 199: what is "functional affinity"? Is there any other?

P6 line 208 "in cellulose" – is that a thing?

P8 line 275 and 280. I fail to understand the relevance of the Thr modification, and specifically how this single amino acid aligns with anything (line 280). Maybe this just needs to be explained better. Currently, from what is written here, I am not sure that Gal-GalNAc is typically directly attached to Thr (and even if it is, that Thr would be part of a peptide chain, and not a free amino acid, right?)

P9 line 308-316: First of all, this paragraph should probably be moved to the discussion section. I am also not happy with the explanation of mechanistic detail here – it is all a bit convoluted and does not refer to the figure (that the authors provide later in the discussion).

Discussion

P9 line 319-331 – this section would benefit from comparison with other autotransporters from different bacterial species, rather than a collection of toxins

P10 line 344: why "due to"? Similarly, why in line 346: "in contrast"? The causalities here could be described/explained better. It is not self-evident that multivalent interactions are based on low affinity, for example.

P10 line 350 please add the affinity values (from literature) here in the text, it would be interesting to compare these to the ones measured in this work.

Methods section:

Please add a section/table with all cloning primers etc used for this study (probably in the section after line 388)

P 11 line 382: custom synthesized DNA (not a "gene sequence") was purchased here.

P11 line 401 and following: what antibiotics were added to the overnight culture?

P13 line 451: what plasmids/constructs were used for expression of hTIGIT in E.coli?

P17 line 640: "to a McFarland standard 3" sounds weird, please rephrase. You did not use a standard here.

P18 line 641: it is not entirely obvious why Jurkat cells are added here to grow Fn. I guess this sentence is misplaced and should be part of the section on cell binding assays.

Figures:

Figure 1 vs Figure 2: in Figure 2, not all exemplars of adhesin fibers are kinked. From Figure 1, the reader could get the impression that the kink is a "given", while from figure 2, one could assume that there are multiple possible conformations. This is not really discussed in the manuscript anywhere? See also supplementary figure 2 where again different conformations are visible.

Reviewer #3

(Remarks to the Author)

The work brings interesting integration between experimental techniques, such as cryogenic electron microscopy (cryo-EM), and computational ones, such as docking and molecular dynamics (MD). The structural analysis of the extracellular domain of the Fap2 adhesin involved cryo-EM, single particle analysis, structural prediction by AI using AlphaFold2, docking and MD, obtaining a 46.5 nm long β -helix. Parameters used for the docking simulations are shown in Supplementary Table 3 and docking and molecular dynamics protocols are detailed in the methods section. However there is little information about the protocol used with AlphaFold2, the authors could extend this topic in the methods section, mentioning more details than the prediction of large proteins piecewise.

Another question regarding protocols refers to that used in MD simulations. Several 50 ns replicates were performed for the Fap2-ECD/hTIGIT-ECD and Fap2-ECD/Gal-GalNAc-Thr complexes. The authors could justify the strategy used, considering which would be the best of two scenarios: i) if for the initial structures in each replica different poses obtained from the docking calculations are used or ii) if the same pose obtained from the docking results is used with different initial atomic velocities. Which could be more statistically relevant? Regarding the lengths of the trajectories, would 50 nsec be enough to scan the conformational space to observe the stability of the complexes and be confident about the molecular

interactions observed?

Reviewer #4

(Remarks to the Author)

In this manuscript, Schoepf and colleagues provide structural, and functional analysis of the outer membrane adhesin Fap2 present on the surface of the oral opportunistic pathogen *Fusobacterium nucleatum*, using a heterologous system in *Escherichia coli*. Through cryo-electron microscopy, structure prediction, and modeling, the authors demonstrated that the rod-shaped extracellular region of Fap2 binds to TIGIT, a T-cell protein, and identified a binding pit on the tip of Fap2 for Gal-GalNAc present on cancer cells. Overall, the findings are significant, greatly contributing to understanding of the molecular interactions between *F. nucleatum* and host cells, and the manuscript is well written. However, there are a couple of major issues that need to be addressed for further consideration.

Major comments:

- 1) While *E. coli* provides a versatile experimental system for this work, it cannot simply replace the native organism *F. nucleatum*, given that genetic tools have been developed for this microbe. There might be potential post-translational modifications in *F. nucleatum* that might affect binding and not be found in this heterologous system. To address this important issue, the authors should perform comparative mass spectrometry analysis (MS/MS) of Fap2 proteins isolated from *E. coli* and native *F. nucleatum* since both are available.
- 2) The authors should confirm the binding site of Fap2 for Gal-GalNAc by mutational analysis since the binding pit has been identified.

Minor comments:

- 1) Introduction or relevant sections: more background information on the AIDA autotransporter systems would be helpful.
- 2) Discussion: The authors consider elaborating more on the biological significance of the findings. How are the observed molecular interactions between Fap2 and receptor molecules related to the broader context of fusobacterial roles in cancer and other physiological processes?
- 3) Page 3, line 83: change "46,5" to 46.5" and check the manuscript throughout.
- 4) Page 3, line 85: please spell out "MD" as first appears.
- 5) Page 5, line 149: "The triangular extensions at the matchstick region result in two pits at the tip (Fig. 1J), which are only present in Fap2, but not RadD (SI Fig. 1A)". It is difficult for this reviewer to make this distinction from the secondary structure of RadD alone in SI Fig. 1a alone. Can the authors indicate what two pits are referred to in Fig. 1j?
- 6) Page 9, line 314: consider revising to "...keep active NK cells a safe distance from Fn itself".
- 7) It may be helpful to label N/N-termini in Fap2 ECD (Fig. 1g).
- 8) Italicize *E. coli* in SI Fig. 2b and check the manuscript throughout.

Reviewer #5

(Remarks to the Author)

Critique 1:

This manuscript presents the cryoEM structures of the adhesin Fap2 and its complex with the T-cell receptor. Fap2 is a pivotal pathogenic factor of Gram-negative *Fusobacterium nucleatum* (Fn). Fap2 extracellular domain (Fap2-ECD), a 400 kDa autotransporter protein, plays a crucial role in binding to cancer and immune cells via two receptors: the glycan Gal-GalNAc and the T-cell protein TIGIT. Given its significant presence in the intestinal microbiome, associated with colon cancer, the third most prevalent cancer globally, Fap2 emerges as an important drug target.

The cryoEM map Fap2 shown in this manuscript is seen as a relatively unique "matchstick" structure—a 50 nm long, rod-shaped molecule with a receptor binding tip. The cryoEM map, resolved at 4.7 Å, fell short of resolving the beta strands, precluding de novo model building. Utilizing AlphaFold2, they proposed a model of a beta-helix with a 1010 kink to fit the density map. Additionally, they resolved the cryoEM structure of the Fap2-ECD complexed with human glycosylated TIGIT-ECD at 6 Å, highlighting the receptor binding site on the distal tip of Fap2ECD through a difference map. Due to the moderate resolution limiting the resolvability of side-chain interactions, they used molecular dynamics to aid in deciphering interactions between Fap2 and TIGIT.

The strengths of this manuscript are the successful expression of the largest membrane-associated protein in *E. coli* and the attainment of subnanometer resolution cryoEM maps. In addition, they used cryoET of cells overexpressing Fap2 to show the matchstick shape of Fap2 to confirm the authenticity of their structural observation of biochemically purified protein. AlphaFold2 and MD simulation were used to build models in elucidating the structure and functional mechanisms governing Fap2's interactions with tumor and immune cells. However, several weaknesses appear in the manuscript:

1. Admittedly, reporting cryoEM structures in this moderate resolution zone poses challenges for constructing accurate models. Lacking rigorous validation such as a PDB validation report to assess reliability and accuracy places uncertainty on the reporting models.
2. Detailed residue interactions between Fap2 and the two receptors in Figure 4 rely on docking and MD simulations. I am not convinced that this method of analysis alone can be trusted. Incorporating additional biochemical experimental data or

improving the map resolutions would be necessary to support the hypothetical models.

In view of the uncertain nature of the models and chemical interactions, the proposed chemical mechanism seems to be somewhat speculative.

Critique 2:

The authors investigated Fap2, a *Fusobacterium nucleatum* (Fn) adhesin, and its interaction with the glycan Gal-GalNAc and a T-cell protein TIGIT. This is important for understanding the role of Fn in the progression of colorectal cancer.

The authors first recombinantly express Fap2 on the E.coli surface and purify Fap2-ECD. A 4-5 Å resolution map was obtained using single-particle cryo-EM analysis. An alphafold2 predicted model was used to fit into the cryo-EM density. A Cryo-ET of native Fn revealed Fap2's native conformation and confirmed that the purified Fap2 has a similar conformation as that of the native Fap2. A fluorescent Gal-GalNAc was used to show binding to Fap2-expressing E.coli as well as fluorescent micrographs showing Fap2-expressing E.coli binds to cancer cells. SPR was used to confirm purified TIGIT binds to purified Fap2. Finally, the authors also obtained a 6 Å cryo-EM map of Fap2-TIGIT and docked it with the known TIGIT structure. Gal-GalNAc was also docked to the Fap2 model to predict its binding site. MD simulations of both Gal-GalNAc and TIGIT interactions further support the stable bindings.

Comments

● Major

1. The cryo-EM maps are in the 4-6 Å resolution range. Although the model building would be challenging at this resolution, the alphafold2 predicted structure fits relatively well visually. However, it is difficult to judge the quality of the model fitting only by looking at the figures. A metric such as Q-score or other correlation-based methods might be useful to assess the quality of the fitting.
2. It would be nice to demonstrate the quality of the map and model fitting by zooming into some local area such as beta sheets or the N terminal tip.
3. Supp fig. 4: Why does it require a density map generated from the alphafold2 model as an initial model for non-uniform refinement? A 5 Å filter is not sufficient to prevent model bias. In addition, this step is probably not necessary as the map (4.79 Å, 265,956 ptcls) is visually similar and with a similar resolution as the map without the generated initial map (4.73 Å, 265,956 ptcls).

● Minor

1. Line 131 and 135: Not clear what "4,5 nm" and "1,2 nm" mean.
2. Line 138: Indicate where the "two triangular extensions" are in Fig. 1g.
3. Line 139: Indicate where are the residues K525-K836 in Fig. 1h.
4. "Longer branch", "tip of the longer branch", and "matchstick region" were used randomly in the manuscript. It's not clear what they mean. Please clarify and use them consistently.
5. Fig. 1d should color the amphipathic helix and unstructured N-terminal region similar to that in Fig. 1b to make it clear that the two regions are in the construct.

Reviewer #6

(Remarks to the Author)

Version 1:

Reviewer comments:

Reviewer #1

(Remarks to the Author)

Thanks a lot for providing a detailed reply addressing all the points raised during the review.

I suggest that you either replace the SDS-PAGE shown in Fig1 with the gradient gel from the reply or you add the gradient gel as supplementary data.

Very nice work!

Michael Kolbe

Reviewer #2

(Remarks to the Author)

all of my comments have been addressed. The reviewed version reads well.

Reviewer #5

(Remarks to the Author)

The authors have made significant improvements in the revised manuscript, enhancing readability and presenting their findings more clearly. In this version, they provide additional evidence supporting their original hypothesis that *Fusobacterium nucleatum* adhesin Fap2 binds to Gal-GalNAc on colorectal cancer (CRC) cells and T-cell protein TIGIT, contributing to colorectal cancer progression.

Notably, the authors generated an additional mutant to abolish Gal-GalNAc binding, guided by their docking model. The results convincingly demonstrate that the predicted interactions are critical for binding. However, the mutant designed to disrupt hTIGIT binding did not validate the predicted interactions, leaving the precise nature of the Fap2-hTIGIT interaction unresolved.

The authors have addressed all my comments from the first round of review. However, I have a few additional comments based on the revised manuscript:

Comments:

Could the authors clarify why a negative control with *E. coli* (similar to Figure 3b) was not included in Figure 4e? This would help confirm if both WT and mutant bind hTIGIT.

Given the negative results in Figure 4e and Supplementary Figure 9a, along with the low resolution (~ 12 Å) of the cryo-EM map at the hTIGIT interaction region, the docking-based interaction model in Figure 4d should be interpreted with caution. The authors may consider moderating their description (Lines 271-287) to reflect these uncertainties.

Could the authors provide a cryo-EM map movie for the Fap2/TIGIT complex, similar to the one provided for Fap2 alone? This would allow readers to better assess the map quality and set appropriate expectations for the accuracy of the predicted interactions in Figure 4d.

Revision for Schoepf F. et al.: Structural basis of Fusobacterium nucleatum adhesin Fap2 interaction with receptors on cancer and immune cells

We thank all six reviewers for their helpful and constructive feedback, which helped us to improve the manuscript. Please find our detailed answers point-by-point below, highlighted in blue font. The changes passages in the manuscript and in the SI were also highlighted in blue.

Reviewer #1 (Remarks to the Author):

Review Structural basis of Fusobacterium nucleatum adhesin Fap2 interaction with receptors on 2 cancer and immune cells

In the current study Roderer and colleagues provided the structure function analysis of the adhesin Fap2 from the Gram negative Fusobacterium nucleatum. They purified the adhesin after heterologous expression in E. Coli, tested its interaction with CRC receptor glycan Gal-GalNAc and T-cell marker hTIGIT. The authors used cryoEM in combination with fluorescence microscopy and other spectroscopic methods. Based on their results they proposed a model for the role of Fap2 in bacteria binding to tumors and deactivation of responding NK cells

Overall the manuscript reads quite interesting and the results deserve publication in Nature Communications. There is however some major and minor issues that need to be solved before acceptance of the manuscript.

Major concerns

My first major comment is regarding the data presented in Fig3. Experiments shown in this Fig show necessary results. For their argumentation it is necessary to show the co-localization of the ligand with the Fap2 using two different fluorescent markers

Although the suggested co-localization experiment would be compelling to carry out to see that recombinant Fap2 on the E. coli surface would indeed specifically bind to Gal-GalNAc on the cancer cells, the glycan receptor limits the practicality of a 2-label super-resolution fluorescence microscopy experiment. Gal-GalNAc is an O-glycosylation of many different glycoproteins in the CRC cell membrane, and therefore labeling specifically one particular O-glycosylated protein using a fluorescence-coupled antibody would not be feasible. If we would label Gal-GalNAc instead with a fluorescent lectin (e.g. PNA as used in Abed J. et al., 2016) or a chemical probe, the Fap2 binding would not work anymore. For Fap2, a GFP-tagged construct at the N- or C-terminus (location at the cell membrane surface and inside the cell in the cytoplasm, respectively) would be located more than 50 nm away from the binding site (i.e., no clear co-localization in super resolution microscopy anymore), and a custom antibody against Fap2 would likely interfere with binding.

We acknowledge the limitations of our experiment, which shows that Fap2-expressing E. coli bind to the surface of cancer cells, but cannot demonstrate that specifically Fap2 facilitates the binding to particular glycosylated receptors. Therefore, we rephrased the description and analysis of the experiment to underline that Fap2-expressing E. coli bind to the cancer cells and not specifically Fap2 (2021): “Next, we tested whether the Fap2-expressing E. coli are able to adhere to colon cancer cells. For this, we stained E. coli with a membrane-specific dye and used fluorescence microscopy to study their attachment to the CRC cell line HT-29. Indeed, only Fap2-expressing E. coli attached to the surface of HT-29 in small clusters, whereas E. coli did not...”

We also rephrased the concluding statement of this paragraph (L 215ff) to: “**Our experiments suggest that recombinant Fap2 provides sufficient affinity to allow association of Fap2-expressing *E. coli* to cancer cells, even to those with low Gal-GalNAc surface levels such as HT-29³⁶, ...**”; and we added a final note to this paragraph (L 219ff): “**We note, however, that direct binding of Fap2 to Gal-GalNAc cannot be demonstrated without fluorescence co-localization experiments.**”

My other concern is with regard to the interesting results presented in Fig4. Those structural and modeling results need to be verified by mutational analysis of residues that were found to be specifically involved in the intermolecular interaction

We fully agree with reviewer 1 and created Fap2 mutants targeting the two receptor binding sites: R613P_E619A_F818P in the Gal-GalNAc binding site and K368P_S371A_R427A_W540A in the TIGIT binding site, respectively. Note that the proline mutants were used for residues with main chain interactions as evident from MD simulations and shown in Fig. 4 e,g. These variants were tested for their interaction with Gal-GalNAc and TIGIT using the same experiments as for Fap2 WT.

For the Gal-GalNAc binding site, HT29 cell binding studies revealed that Fap2_R613P_E619A_F818P expressing *E. coli* is impaired when compared to WT (~30% less *E. coli* bind to cells, Fig. 5 c,d). This is now described in the Results section (L 341ff): “**To confirm the key roles of these residues in the pocket for Gal-GalNAc binding, we created the Fap2 triple mutant E619A_R613P_F818P (Ala mutation for side-chain interacting E619, Pro mutations for main-chain interacting R613 and F818), and tested cancer cell adhesion of *E. coli* that express the resulting Fap2-ECD variant. The adhesion of *E. coli* that express Fap2-ECD_E619A_R613P_F818P was indeed slightly impaired, with an average binding signal reduction to ~70% of the wild type (Fig. 5c,d). This difference is significant, but not as pronounced as the reduction in binding caused by the addition of an excess of free GalNAc (Fig. 3b). We therefore conclude that these residues and their proposed H-bonds are important for binding, albeit other residues might provide redundancy. Further high-resolution structural research is required to confirm the exact molecular nature of interaction.**”

For the TIGIT binding site, a quadruple mutant of the predicted Fap2 side chains however did not result in decreased binding intensity to TIGIT-overexpressing HEK293 cells (Fig. 4e,f), and also in pulldown assays (SI Fig. 9a). We now describe and discuss the possibility of redundancy in side chain interactions, and limitation of our approach in the Results section (L 287ff): “**We next created a Fap2-ECD mutant of the binding interface that would remove the identified main-chain and side-chain hydrogen bonds (K368P_S371A_R427A_W540A), expressed it on the *E. coli* surface and tested binding of the bacteria to HEK293 cells that overexpress hTIGIT-eGFP. Unexpectedly no significant differences in hTIGIT-expressing cell binding between *E. coli* that express Fap2-ECD wild type (WT) and this mutant became evident (Fig. 4e,f), with nearly identical mean and average binding values. Furthermore, in a pulldown assay using purified Fap2 WT and the mutant, no qualitative differences in their ability to bind hTIGIT-ECD were observed (SI Fig. 9a). These results are an indication for further possible binding modes of the hTIGIT-ECD loop within the cleft at the Fap2 tip, or redundancy provided by other Fap2 side chains.**” and in L 304ff: “**In summary, our structural assessment of the Fap2-ECD/hTIGIT-ECD complex reveals binding of hTIGIT to Fap2 at the maximally distant position to the Fn surface. Whereas docking and MD simulation suggest a particular mode of interaction via a hTIGIT loop that**

intercalates into a cleft at Fap2, site directed mutagenesis did not confirm the binding model. Further structural analysis of the complex at near-atomic resolution is required to understand the binding mechanism of Fap2 to hTIGIT at its full extent.”

Since the additional mutagenesis data required an expansion of the figures, we decided to split the original Fig. 4 (*Structure of the complex of Fap2-ECD and hTIGIT-ECD and docking of Gal-GalNAc-Thr*) into two figures (4: *Structure of the complex of Fap2-ECD and hTIGIT-ECD* and 5: *Docking of Gal-GalNAc-Thr to Fap2-ECD and validation of binding site*), and to split the corresponding chapter in the Results section into two chapters.

Finally, I was wondering about the obtained resultion for the EM maps presented in the current study. I think the presented date is still structural heterogeneous. The authors should comment if the have fined further “data polishing” by removing noise, and extended 2D classification or if they have considered other strategies.

With the processing strategy outlined in **SI Fig. 4 and 8** and described in the Methods section, we obtained the highest resolution with the best credibility, i.e., with FSC curves without dips, with the map visibly representing the reported resolution, and with the final 3D refinement in Relion revealing identical resolution values (see methods). We indeed carried out several rounds of 2D classification, as indicated in the schemes of **SI Fig. 4 and 8**. Other 3D classification strategies (e.g. different numbers of classes, other masks, heterogeneous refinement, 3D initial model with different numbers of classes) did not result in better maps. Importantly, heterogeneous refinement (or 3D classification with alignment) resulted in shortened reconstructions of the Fap2-ECD (see detailed answer to the 3rd comment of reviewer 5 on p.25) We believe that without a substantially increased size of the data set (~10x more particles) the limitations in resolution will persist. So far, we are unable to achieve high particle density in the holes of the cryo-EM grids, especially for the Fap2/TIGIT complex. Also, graphene grids that we tried for the complex and Fap2-ECD alone did not help to overcome this limitation. We summarize these limitations in the discussion **(1, 4) 7(f)**: **“At this point, our integrated approach is at its limit and high-resolution data of the complex are required to ultimately visualize the involved side chain interactions at the binding site.”**

Minor concerns

Line 78 to 91: Wy too detailed? I appreciate that the authors recap the current situation and define the open questions in the filed. The many details (starting from lin 83 until the end of the introduction) however, should be presented in the discussion and not in the introducing paragraphs.

We thank the reviewer for this suggestion and reduced the level of details in the last paragraph of the introduction. The paragraph (**1, 72(f)**) now reads as:

“We have therefore developed an expression strategy for the recombinant production of Fap2 from *F. nucleatum subsp. nucleatum* ATCC23726 on the OM surface of *E. coli*, which resulted in a functional Fap2 extracellular domain (ECD) that binds to both receptors, Gal-GalNAc and TIGIT. Structural analysis of the Fap2-ECD by cryogenic electron microscopy (cryo-EM) in combination with AlphaFold2 structure prediction revealed a 46.5 nm long rod-shaped β -helix with a matchstick-like tip that interacts with the immune cell receptor TIGIT and the CRC cell receptor Gal-GalNAc at two different sites.”

Line 94: title does not reflect the content of this chapter

We changed the heading of this chapter (now **8**) to “**The Fap2-ECD forms a rod-shaped kinked β -helix.**”

Line 114-16: authors are not precise here. Moreover, showing an alignment of two orthologous genes to argue about conservation aspects is not convincing. Authors might add more sequences to the alignment or they delete this comment.

We now specifically point out that we compare the Fap2 sequences of two Fn “WT” strains for which the sequences of Fap2 are known (the text in the Results section and legend of Supp. Fig.1). We also replaced the too imprecise statement of conservation by the notion that the parts of the sequences are highly similar in the two Fap2 (**100%**): “**While the sequence parts corresponding to autotransporter, the linkers, and the β -helix are highly similar in Fap2 of two Fn strains, more variations are evident at the N-terminal part between Fap2 of Fn ATCC23726 and Fn ATCC25586 (SI Fig. 1b).**”

fig S1: How does the alignment in panel B relate to the structures shown in A? Labelling of the region including residues from 300 to 1000 would be great

We thank the reviewer for this suggestion and highlighted the region of the alignment in panel a in the two Fap2 models in **SI Fig. 1**

Line 126: and Fig1 do not match. Apparent molecular mass can differ sometimes a bit but the observed difference of 50 to 60kDa seems very high. The authors should verify the length of the purified protein by tryptic digestions and mass spectrometry or another method that reveals sequence details

We respectfully disagree with reviewer 1 on this point. The resolvability on SDS-PAGE is low on the applied gel (7.5% acrylamide) in the region of around 300 kDa, therefore, differences of 50 kDa are not well seen. For another purification, we used a gradient gel, where the Fap2 ECD migrated above the 300 kDa band (see below).

The N-terminal region is visible in the cryo-EM density (i.e., the unstructured region complementing the hydrophobic groove) and the C-terminus with the histag has to be present

for purification; therefore we can safely deduce that the analyzed Fap2 ECD has the expected length.

Line 140: the authors do not give any details on the quality of the obtained fitting results. Add please.

We added the following information (L 136ff and SI Fig. 3f): **“The structural model was then adjusted by a molecular dynamics (MD) simulation based fit in Namdinator⁴⁶, followed by manual model adjustment in Isolde⁴⁷ and real-space refinement in Phenix⁴⁸ (Fig. 1h) Although the final map-to-model FSC value was limited to 6.5 Å (SI Fig. 3i), likely because of the non-separability of the β-strands in the β-helix, the data together with the model allowed interpretation of several unique key features of the structure.”**

Line 140-43: I cannot follow the authors argumentation checking the panels I and J of Fig1. The authors should explain why in panel I two different model fittings are presented to the reader and how does this relate to panel J. Taken together, panel G to J are a bit redundand and the point with the hydrophobic groove is poorly illustrated. Some of the panels could go into supplement.

The two models in Fig.1i illustrate the longitudinal hydrophobic groove that is closed by the unstructured proline-rich N-terminal region (195-361). For this, the right illustration shows a surface model without the N-terminal part. We recognize that we had an error in the reference in the text (Fig.1h instead of Fig.1i) and corrected this, and in addition added references to the left and right panels of Fig.1i separately. We also added the following sentence to the Fig.1 legend: **“Note the exposed hydrophobicity without the N-terminus (right).”**, and added boxes, arrows, and zoom-ins to the illustration of Fig. 1i to highlight the “open” hydrophobic groove in the right panel. We furthermore recognize that the information of Fig. 1j (surface representation of Fap2 model colored by electrostatic potential) is misplaced here and moved it to SI Fig. 1 (as panel c) to illustrate the pits on the Fap2 matchstick head, and also comparing with the predicted model of RadD that has no pits (see also comment of reviewer 4). This is now referred to in L 152. Due to redundance, a similar representation in SI Fig. 10 (previously; SI Fig. 9) was removed (reference in L 329 changed to SI Fig. 1c accordingly).

Line 149-50: This comparison is invalid. Fig1 J is showing an EM map while SupplFig1 shows an alphaFold model

Fig. 1J indeed shows a surface representation model, not an EM density map. We added this information to the figure legend (now in SI Fig. 1).

Line 152: change the text to
putative receptor-binding β-helical ECD

We thank the reviewer to point this out and changed accordingly (now in L 155).

Line 169-70: The given argumentation is not convincing in light of Fig2 panel C. Indeed, some structures match while other do not. Did the authors tried comparison of the observed cryoET species with SPA 2D class averages?

We recognize that our argumentation is an overstatement and we toned it down to (L 171ff): **“Moreover, the membrane-distal ends of some cell attachments indicate thickenings, in**

agreement with the longer part of Fap2-ECD". The rod-shaped structures on the F. nucleatum surface do not only relate to Fap2, but also to other proposedly similarly shaped adhesins, especially RadD, which also has been identified in the OM preparation (Fig.2a, SI Fig. 5a,b). Therefore, not all Fn surface structures can match the Fap2 structure. We now clearly point this out in the manuscript (L. 176f): **"We however note that at the resolution level of a cryo tomogram we cannot distinguish between Fap2 and RadD because both both appear to have the same length and a very similar overall shape"**. Moreover, we now also show comparisons with 2D class averages in Fig. 2c to highlight that orientation differences relative to the viewing axis result in visually longer and shorter attachments, and added this information to the text (L. 180): **"... and different orientations alongside the membrane..."**.

Line 170-72: The negatively stained specimen shown in SI Fig. 5e are not conclusive. The negative stain images can only illustrate the overall shape of the molecules, i.e., rods that protrude from detergent micelles (in which the C-terminal autotransporters are embedded). Only together with the SDS-PAGE we draw the conclusion that the cell attachments (i.e., autotransporter adhesins) retain their shape and integrity. We recognize that the presentation of the NS-EM data require improvement here, and therefore we include a magnified inset together with a distance measurement of the micelle-embedded rods. We also added more details to the descriptive sentence in the text (L. 174f): **"... as indicated by SDS-PAGE in which no degradation is evident and negative stain EM in which rod-shaped kinked molecules of up to 50 nm length are visible"**

Line 187-89: I do not understand what is shown here. The authors might revise the plot shown in panel A. What is the control in this experiments, how were the data fitted, how was the affinity calculated? Explain and show data, please

We recognize that details of the plot in the Methods section are not sufficient, and we provide further information now in the legend of Fig.3a: **"Data points represent individual measurements per concentration and dashed lines indicate the mean values. Data analysis and visualization was carried out in R using the ggplot2 package⁹⁵."** We also revised the plot in such that we now show Gal-GalNAc binding to control *E. coli* and Fap2-expressing *E. coli* in the same plot. We point out that no affinity was calculated here, only a descriptive analysis when a level of binding saturation was reached. We furthermore acknowledge that the direct comparison of our data with cell and tissue binding data from Abed et al (2016) using *Fusobacterium nucleatum* and not Fap2-expressing *E. coli* and soluble Gal-GalNAc is misleading. Therefore, we rephrased the text to (L. 201f): **"In fact, the difference between Gal-GalNAc binding for Fap2-expressing *E. coli* and *E. coli* expressing pAIDA without Fap2 increased until a plateau at Gal-GalNAc concentrations of 50 - 100 μ M is reached (Fig. 3a). This means that recombinant Fap2 binds to the cancer cell receptor at least as well as native Fap2 on Fn, where cellular and tissue binding studies were carried out with Gal-GalNAc concentrations in the mM range."**

Line 191-94: The argumentation and the corresponding data presented in Fig3 look nice. However, the authors should consider co-localization studies using an additional fluorescent marker for Fap2 or they have to tone down their argumentation a bit. Please see our answer to the first major comment above. Because super-resolution co-

localization studies are not doable for us with Fap2 and Gal-GalNAc, we toned down our statements (L2150, L2190).

Line 216-24: I have problems understanding the co-purification illustrated in SFig7. In corresponding panels a to c the hTIGIT construct has a molecular mass of about 40kDa while in the pull-down experiments shown in panel d the mass is almost 20% higher. The authors should explain this mass difference

In panels a and c, the hTIGIT-ECD-GFP sample has been loaded on the SDS gel without previous heating in order to leave the GFP tag intact for in-gel fluorescence micrographs. In panel d, the sample has been heated to 95°C. We now provide the information in the figure legend after panel d.: **“Note that the apparent size differences of hTIGIT-ECD-GFP in SDS-PAGE originate from the fact that protein samples were heated in d and not heated in a and c”.**

Line 229-31: SPR spectra seem to be sub-optimal (baseline shift, bulk shift which could have several reasons. Moreover the authors do not show here or elsewhere in the supplement the Ru as a function of concentration. This data is essential to verify the quality of the SPR experiments

The **SI Table 3** (previously SI Table 2) shows the fitted RU, k_a , and k_d values including their errors, which is referenced in the text. Following the reviewer’s suggestion, we added a plot of the RU values as a function of concentration (**SI Fig. 7d**). The limited solubility of Fap2 prevents to use higher concentrations in the experiment, thereby saturation in the concentration-dependent RU plot was not reached. To cope with this limitation, we now specify the K_D with one valid digit (i.e., 0.6 μ M instead of 580 nM).

Line 243-50: Check my comments on Fig. 4 (see below)

I see two questions that need to be addressed by the authors.

Is there an additional density at the tip/

is there further structural differences when hTIGIT binds?

The later questions was not at all addressed. The authors should comment on this issue.

Panel a to d of Fig4 is pretty redundant. One panel in main fig. Might be enough. Rest of the data can go to the supplement.

The signal subtraction (**Fig. 4c**) shows the additional density at the tip. No further comparably large changes are evident throughout the β -helix. We added this information, including **SI Fig. 8c** that shows the entire Fap2 ECD density with difference density (**L 2700**): **“No other difference densities are seen outside of the Fap2 structure, indicating that TIGIT binds only at the tip and that binding does not induce structural changes elsewhere in Fap2 that are apparent at this level of resolution.”**

Although Fig. 4 panels a-d are redundant to some extent, they are in our opinion useful to (a) illustrate the full structure, (b) illustrate shape of tip outside Fap2 model, (c) illustrate perfectly the additional density at the tip, and (d) show the complex model obtained by integrative modeling. As we now show the entire ECD with additional density in **SI Fig. 8c**, we removed Fig. 4c from the main figures. We prefer to keep the other three panels together in Fig.4 to visualize the information at a glance and not distributed over several figures.

We acknowledge that our description in the text was convoluted, we therefore change the passage at **L 2580** to: **“We found that at the tip of the matchstick, density appeared fragmented likely because of a mixture of Fap2 with or without bound hTIGIT. We**

therefore proceeded with focused 3D classification on the tip and combined subsets with clearly defined density beyond the Fap2 matchstick, resulting in 45,207 out of 73,482 particles for final refinement (SI Fig. 8c,d). The resulting density map with a global resolution of 6.0 Å (SI Fig. 8) revealed a long, kinked, rod-like structure, similar to the Fap2-ECD alone (Fig. 4a). In the complex, the tip of the longer part of the β -helical rod has an additional, angular-shaped density, whereas the tip of Fap2 alone appears round-shaped (Fig. 4b).”

Fig1: color code is confusing: panel B and D use similar color for different regions

Too many different font (& size) used

panel B: linker is not part of the passenger domain

panel I and J: what is the point with the hydrophobicity/hydrophilicity? This figure is not helpful to follow the authors argumentation

We thank the reviewer for pointing this out and changed font and size to be consistent (headings – Arial 7 pt, descriptions – Arial 6 pt). We also changed the marking of passenger domain in b, and adjusted color code in d (passenger domain, other shades of colors for the E. coli signal sequence and autotransporter). We moved panel j to SI Fig. 3 (see also my answer above to line 140-143 regarding the relevance of panels i and j).

Fig2. Panel B: OM looks corrupted. The authors might comment on this

Visual damages of the OM of the obligate anaerobe *F. nucleatum* might originate from exposure to oxygen before vitrification. We added this information to the legend of Fig. 2: “**Note that the ripples in the OM might be caused by oxygen exposure of the anaerobic *F. nucleatum* before and during vitrification.**”

Fig5: Not conclusive: Does that imply that FN simply forms a distance barrier?

Indeed, one important aspect of our model is that Fn applies Fap2 to keep NK cells (and T-cells) on a distance from itself and the colonizing tumor, as written in the last paragraph of the results section and in the discussion (spear and grappling hook hypothesis). We added this illustrative analogy also to the Fig. 5 legend.

FigS1: label N- and C-terminus, correlate panels A and B

We labeled the termini as suggested and indicated the region of the alignment (panel b) in panel a.

Reviewer #2 (Remarks to the Author)

The manuscript «Structural basis of *Fusobacterium nucleatum* adhesin Fap2 interaction with receptors on cancer and immune cells» by Schoepf et al. describes structural and functional details of Fap2 binding to its two main receptors, one a signaling receptor of cancer cells, and the other a carbohydrate component of the extracellular matrix. The manuscript is generally well-written, and I have only a few major comments (and minor comments mostly relating to language)

Major comments:

1) In different places in the text, the authors suggest directly or indirectly that they are the first to purify a Fusobacterial autotransporter, and/or neglect to mention that parts of their methodology has been used successfully previously in related experimental systems. I am not suggesting that this is deliberate (it's oversights – it is a complex field, and purification strategies are not always obvious from abstracts); nevertheless, I suggest to modify some claims made related to the protein purification strategy, and to amend some text with additional citations:

We thank the reviewer for pointing this out to us, and we recognize that our claims were too unprecise. It was indeed not deliberate to neglect the previous work in this field. Our work claims the first recombinant expression and purification of large Type Va autotransporters (such as Fap2, RadD, Aim1) of *F. nucleatum*, which, to our knowledge, has not been reported so far. We therefore amended changes and added citations, as outlined here:

- **Fig. 72f**: shortened last paragraph of introduction according to suggestion by reviewer 1, which removed a sentence that could imply we were the first and only ones to produce fusobacterial autotransporters. In the first sentence of this new paragraph, we write “expression strategy” instead of “expression system”.
- abstract: changed last sentence (see below)
- **Fig. 1d**: We replaced “design” by “schematic overview”
- Discussion: “Fn Fap2-ECD in this work is, to our knowledge, the first membrane-associated protein of such a large size that has been produced in *E. coli* using the AIDA autotransporter system.” → This statement is true, to our knowledge. Therefore, we leave it.
- Discussion: “This has previously only been used to display much smaller “passenger domains” on the *E. coli* surface (Lattemann et al., 2000; Jarmander et al., 2012), while our design proves its general applicability for the preparative production of surface proteins independent of their size and behavior in the *E. coli* cytoplasm”. → We now agree that this sentence, especially the 2nd part, is an overstatement and might take away credit from earlier work. Therefore, we rephrased the entire section (with citations, **Fig. 377ff**): **“Previous work used the AIDA-I-based and other autodisplay system for cell surface display of a variety of proteins and applications^{42,43,46–48}, including cleavable versions^{43,48,49}. Fn Fap2-ECD in this work is, to our knowledge, the first membrane-associated protein of such a large size that has been produced in *E. coli* using the AIDA-I autodisplay system, thereby underlining the broad applicability thereof.”**
- Abstract p1 line 33: “first purification”. A Google Scholar search with the terms “purification of fusobacterial autotransporter” immediately gives a long list of fusobacterial

autotransporters that have been purified for different purposes before. I suggest to simply delete the complete last sentence of the abstract.

Again, we apologize for our unprecise statement here and are now aware of successful recombinant purification of other *F. nucleatum* autotransporters like CbpF (Shhadeh et al, 2021) or FplA (Casasanta 2017), yet these are no large typeVa autotransporter proteins. Therefore, changed the last sentence in the abstract to: **“Our data report the first high-resolution structural analysis of a Fn Type Va autotransporter adhesin and its receptor association.”**

- Introduction p3 line 88-91. AIDA-I based expression systems have been around for a long time, also for expressing heterologous proteins on the *E.coli* surface (people used to call this “autodisplay” in literature. Check “AIDA-I heterologous expression” in Google Scholar). Likewise, proteolytic cleavage of passenger domains of autotransporters is not an original invention of the authors. A significant number of bacterial autotransporters are autocatalytic proteases that release their passenger domain by default (might be worth mentioning). For an example of an autotransporter with an engineered protease cleavage site, see f.ex. www.nature.com/articles/srep28020 ... I suggest to rephrase “our system” to maybe “our strategy” and maybe to add more information on prior work.

Please see our answer above on rephrasing this paragraph. More references to previous work on autodisplay (Jose and Meyer, 2008; Nicchi et al., 2021, Fleetwood et al., 2014, Yoshimoto et al, 2016) were added besides the existing two citations of Lattemann et al. and Jarmander et al. to a revised sentence in the first paragraph of the Discussion (l. 377ff, see above) to highlight a broad application range for the AIDA autotransporter system. More citations of original work (Benz & Schmidt, 1989, Maurer et al., 1997, were added to the first paragraph of the Results section (l. 93).

- Results p4 line 117: here, the autodisplay system using AIDA-I is mentioned for the first time. The paragraph should reference to the original work where this was developed – not only to give credit to the developers, but also to explain better what is meant with “are designed to enable translocation of heterologous cargoes...”

We mention the AIDA-I system already a few sentences above (l. 92 in the revised version) and added citations to original work there, and more in the discussion (see above).

“...are designed to enable translocation of heterologous cargoes...” – this formulation is indeed misunderstandable, we meant that we deliberately did not add the Fap2 region predicted as “linker” (Fig. 1c, res. 3272-3471) and not the Fap2 signal peptide because these two might interfere with the functionality of the pAIDA system. Moreover, we recognize that our formulations might take credit from the original authors. We therefore rephrased (l. 105ff): **“The pAIDA1 plasmid contains the AIDA-I signal sequence, a tobacco etch virus (TEV) protease cleavage site, a Myc-tag and a ~5 kDa linker derived from the AIDA-I passenger domain ahead of the AIDA-I autotransporter^{43,45}. Therefore, we omitted the Fap2 signal sequence and the proposed Fap2 linker between the β -helix and autotransporter (res. 3272-3471) and cloned only the variable N-terminal part and the entire β -helix (res. 42-3271) between the pAIDA signal sequence and the TEV site. To facilitate the isolation of pure Fap2-ECD, we introduced an octahistidine His-tag at its C-terminus and named the resulting plasmid pAIDA_TEV_Fap2EC.”**

- Somewhat related to this: P5 line 173-175: “...our findings suggest that a large number of Fap2 and RadD molecules are present in the Fn OM, where the linker domains allow flexible attachment”. The surface density of these two molecules was most probably quantified by others previously, and if so, this needs citations (and then the findings “confirm”, not “suggest”).

While in previous work (Kaplan et al., 2010) the surface density of Fap2 and RadD has been compared (which we reproduced in Fig. 2a) and mRNA has been quantified, we are, to our knowledge, indeed the first ones to visually display the large type Va autotransporter adhesins as surface extensions by cryo-ET. We rephrased to (L 179f): **“Taken together, our findings confirm previous work that a large number of Fap2 and RadD molecules are present in the Fn OM³⁹, and different orientations alongside the membrane suggest that the Fap2 linker (res. 3272-3471) connects the Fap2-ECD in a flexible manner to the autotransporter domain.”**

As a separate point, it might be good to repeat more clearly here what is defined as the linker domain, and what experimental data the authors mean when they claim that their results are in line with “flexible attachment” (and I assume they mean attachment to host cells, this could be made more explicit too).

Please see our statement and changed text above (L 105f). We meant flexibility between the Fap2 β -helix ECD and the C-autotransporter domain through the linker region.

- Discussion P9 line 319-331 – this section would benefit from comparison with surface display of other autotransporters from different bacterial species (with citations... see above), rather than a collection of toxins. Also, I do not see the point of the size discussion here. I am not aware of any size limit for the autodisplay systems (using AIDA-I or any other autotransporter). Proteins of any size should potentially be “expressible/exportable” through these systems. Known limitations are charge, premature/fast folding, and potentially, proline residues or other sequence features that might hinder export through a narrow pore.

Our previous experience (i.e., work with large bacterial toxins, attempts to produce Fap2 in the *E. coli* cytoplasm) clearly underlined that it is not trivial to produce large proteins in *E. coli*, therefore we consider this section in the discussion significant. Below we show an SDS gel of a Fap2 expression and purification attempt in the *E. coli* cytoplasm in which many smaller bands (“smear”) below the Fap2 main band (slightly above 300 kDa) indicate proteolysis during expression and/or incomplete expression.

SDS-PAGE showing expression and purification of Fap2 extracellular domain in the cytoplasm of *E. coli* using pET28a and *E. coli* BL21DE3 STAR. E1 – E8: elution fractions from NiNTA chromatography. The band above 300 kDa corresponds to full-length Fap2, whereas the “smear” between 250 and 100 kDa co-localizes with the Fap2 elution.

With our reformulations of the second half of the paragraph (see above, [377]), we bring in more citations to mention a broad applicability of autodisplay systems.

2) I find the binding of Fap2 to two different receptors interesting, but I do think that some of the data needs better controls (with minimal effort and using methodology the authors already employ, see below). In detail, what I mean is:

- P5 line 182-183 (and later P6 line 216): here the authors themselves suggest that Gal-GalNAc is part of complex glycosylation patterns (true). They later state that hTIGIT is glycosylated (see below). What I am really missing is both a discussion, and experimental controls that the binding that is observed to Gal-GalNAc and to hTIGIT is possibly the same thing. Or can the authors be very sure that glycosylated hTUGIT does *not* have a Gal-GalNAc motif in its glycans?

In the colon, Gal-GalNAc is exclusively present as O-glycosylation cancer cells (functions even as biomarker for colon cancer) due to mutations in the glycan maturation system, which stops glycosylation on the Core-1 (Gal-GalNAc) glycan structure. TIGIT, an immune cell glycoprotein with 2 N-glycan sites is not present on cancer cells. Therefore, TIGIT does not have the Gal-GalNAc glycan. At the mentioning of TIGIT (originally [216], now [235]), we now add that TIGIT has 2 N-glycosylation sites: **“To study the interaction of Fap2 with native-like, human TIGIT (hTIGIT) that has two N-glycans on N32 and N101...”** The O-glycosylation was already stated for Gal-GalNAc in [197]. We also moved the citation (Lin et al., 2021) that describes the TIGIT glycosylation to the end of the new sentence.

Based on the above, I am really curious why the authors did not do the obvious control experiment: a competition experiment where they “interfere” with Fap2-hTIGIT binding in the presence of increasing amounts of Gal-GalNAc. This should be done both with pulldowns and SPR, and probably laso in the cell-binding experiments, in the same way they did the original binding studies to hTIGIT (page 6/7). If the authors are correct, then there should be no competition. If there is, then their presumed binding sites are at least overlapping. In any case, this control experiment would confirm (or not) the trustworthiness of the MD/docking studies.

- This obviously then would relate to changes in the discussion, especially around page 10 line 353.

We thank this reviewer for the suggestion. To find out whether the glycan receptor inhibits TIGIT binding of Fap2, we carried out a pull-down of Fap2 and TIGIT in the presence of 25 mM GalNAc. We found that despite the presence of GalNAc Fap2 was still able to bind and pull down hTIGIT, qualitatively without any difference. The data are presented in Fig. 5d and [355]: **“Gal-GalNAc is present as O-glycosylation on cancer cells in the colon³⁷, and the Fap2 immune cell receptor hTIGIT has two N-glycosylation sites that are crucial for its interaction with PVR⁵¹. To assess whether the attached glycan receptor inhibits Fap2 binding to hTIGIT despite the physical separation of the binding sites, we carried out a pulldown assay of Fap2-ECD with hTIGIT-ECD in the presence of 25 mM GalNAc, as used previously in hemagglutination assays³⁶ (i.e., a $\sim 10^3$ fold molar excess over hTIGIT-ECD). We found that Fap2-ECD still bound and pulled down hTIGIT-ECD in the presence of GalNAc, qualitatively without any difference in the band intensities of hTIGIT-ECD and Fap2-ECD (Fig. 5d). This indicates that co-binding of both receptors to Fap2 is generally possible due to physical separation of the two binding pits.”**

Minor comments in order of appearance:

Abstract

P1 line 19: “numerous microbiota”. I am unsure that this is correct terminology. In my view, we are talking about dysbiotic situations here, and not about a subset of healthy interstitial microbiomes/microbiota (microbiota? Is that the correct plural?)

We replaced “microbiota” by microbes, which is the general term and does not imply that *F. nucleatum* and others are constituents of a “healthy” colon microbiome.

Introduction

P2 line 43: the third-most, the second-most... (needs hyphens). See also line 58 CRC-driving...

P2 line 63: “such as by” – poor grammar, please rephrase?

P3 line 76: how is it relevant here that there is no structure of “another” autotransporter from Fn available? Please explain or delete.

We thank the reviewer for pointing out the grammar issues and changed/rephrased them. We also deleted “or another Fn autotransporter adhesin” in l. 76 because this is not relevant for our work.

Results

P3 line 95-104. Nothing in this paragraph is results. This should be part of the introduction I think. I also do not see the value of Figure 1a to the results section of the manuscript – The genomic localization of different Fn autotransporters is not discussed anywhere later, and does not link to any result?

While we agree with reviewer 1 that Fig. 1a is not absolutely necessary to follow the main message of our manuscript, we would like to stress its added value because it shows at a glance the two large type Va autotransporter adhesins Fap2 and RadD besides many other autotransporter proteins in *F. nucleatum*, which are on average larger than other proteins in this bacterium. Therefore, we favor to leave this figure panel in the manuscript. Indeed, this part would also fit well to the introduction, but since we carried out the *F. nucleatum* genome mining analysis for autotransporters including including manual annotation, we deliberately placed it in the beginning of the Results part.

P4 line 127: how was “>90% purity” quantified? Or is this just an “eye-balled” thing based on an SDS gel (then rephrase)?

We added: “as qualitatively estimated by SDS-PAGE” to the sentence.

P4 line 134: “parted in two branches”. I find this expression confusing. “branches” to me suggest a ‘forking’ of sorts (which is not the case here). What the authors are trying to express is that there is a kink in an otherwise pretty linear, rod-like structure. Certainly the ‘kink’ is not the origin of two branches. If this is expression changed here (and I think it should be), make sure that this is also checked in other parts of the manuscript.

We agree that while “kink” is the correct notion, “branches” might be misleading and indicate a "starting point" at the central kink. Following the reviewer’s suggestion, we consistently changed “long/short branch” to “long/short part”.

P4 line 142: “this gap is closed...” – what gap? Do you mean the “groove” from the sentence before?

We indeed meant the groove and changed accordingly.

P4 line 143-145: “The unstructured N-terminus is highly conserved in different fusobacterial species, highlighting also the conservation of the hydrophobic groove,...” – if it has a groove, how can it be unstructured? I am under the impression that this needs better explanation.

The unstructured N-terminus “fills” the longitudinal groove, which is formed by the V-shaped β -helix, and complements otherwise surface-exposed hydrophobicity. To underline that these are two different parts of the protein, we added the amino acid numbers of the β -helix with the groove at this place (amino acid numbers of N-terminus were already present). Moreover, we re-arranged this sentence to be less convoluted. The new sentence now reads as (L145H): **“The unstructured N-terminus is highly conserved in different fusobacterial species and in predicted autotransporter proteins of the *Pseudoleptotrichia goodfellowii* and *Leptotrichia* species that are known human pathogens (SI Fig. 3H), highlighting also the conservation of the hydrophobic groove (residues 393 – 3271 in Fn ATCC23726) that is complemented by the N-terminus. “**

P5 line 151 “our findings report...” findings cannot report.

We thank the reviewer for pointing out this grammar error and changed to: “Our results provide...”.

P5 line 152-154: the authors “conclude” here that their autotransporter acts as a rod-like adhesin “in functional analogy” to other similarly shaped autotransporters. I think this is a bit mis-phrased. First of all, this is probably homology, not analogy. Secondly, it is not really clear to me where the “conclusion” comes from? From comparing shape?

We agree that we cannot “conclude” here, but instead we could “propose”. However, we recognize that this statement in which we conclude/propose analogous function simply through the shape might be overstating. We therefore changed the sentence to: “It has an extended shape similar to other structurally described autotransporter adhesins, such as CdrA/B from *P. aeruginosa*.”

P5 line 164: “both of them being the largest ORFs in Fn (Fig 1a)”. This should probably come with a citation as it is not an analysis performed by the authors. Also, Fig 1a has nothing to do with ORF length?

The analysis in Fig. 1a was done by us capitalizing on fundamentally important work from the Slade lab (Fusoportal). We add the corresponding citations here (Sanders et al, 2018, Umana et al, 2019), but we also leave the reference to Fig. 1a here because it illustrates that the two longest bars (proportional to ORF length) are Fap2 and RadD.

P5 line 171: “the cell attachments keep their shape and identity as indicated by SDS PAGE” – I am not convinced that SDS page can show that? Consider rephrasing.

Reviewer 1 also raised a similar point here, and we rephrased accordingly (L. 174f). We detailed what SDS-PAGE and EM can show us and only results of both experiments together underline our conclusions.

P6 line 188: “pAIDA without Fap2” is presumably pAIDA wildtype (i.e. expresses AIDA-I)?

This is the “empty” pAIDA vector as obtained from Addgene and not AIDA-I WT. We detailed this here (now L. 202f): “...*E. coli* transformed with the “empty” pAIDA without Fap2...”

Same line: “at receptor concentrations as low as 23-100 μ M” - please be more explicit that this refers to the disaccharide (it does, I think).

We added the information that it refers to the disaccharide (... **Gal-GalNAc concentrations of ...**). See also a response to reviewer 1.

P6 line 196/197: “This indicated that surface Gal-GalNAc is not equally accessible to the bacteria within the cell population” – compared to what? And can the authors be sure that surface Gal-GalNAc is evenly distributed on the cells and that this is the reason for uneven distribution of bacteria? I also miss a discussion of the possibility that the bacteria autoaggregate (this is quite common when they express autotransporter adhesins – even though I cannot say for Fap2) and that this is the reason for the observed clustering.

We agree with reviewer 2 that our interpretation is not supported by data and ignores different other possibilities for the observation. Autoaggregation has not been shown for Fap2 on *Fusobacteria*, but it might be possible that other surface proteins on *E. coli* might be Fap2 autoaggregation targets. Therefore, we reformulated it to discuss more possibilities and also switched order with the next sentence (Similar observations were made...) (L. 213ff): “**These data indicate either uneven distribution of Gal-GalNAc on different CRC cells in the population, different surface accessibilities of the glycan to the bacteria, or autoaggregation of Fap2-expressing *E. coli* prior to cell binding.**”.

P6 line 199: what is “functional affinity”? Is there any other?

We removed “functional” before affinity (now L. 216).

P6 line 208 “in cellulose” – is that a thing?

We wrote *in cellulose* to distinguish between *in vitro* (biophysical and biochemical studies of purified proteins) and *in vivo* (in living organisms). However, we agree that in this specific context, “cell-based” system would be the better choice and therefore we now use this expression (now L. 221).

P8 line 275 and 280. I fail to understand the relevance of the Thr modification, and specifically how this single amino acid aligns with anything (line 280). Maybe this just needs to be explained better. Currently, from what is written here, I am not sure that Gal-GalNAc is typically directly attached to Thr (and even if it is, that Thr would be part of a peptide chain, and not a free amino acid, right?)

We decided for several reasons to use the Gal-GalNAc-Thr adduct in the simulations:

(1) availability and suitability in the biochemical experiments (see Fig. 3a), where the Thr enabled us to attach a fluorescent dye to the disaccharide.

(2) natural O-glycosylations occur mostly on Thr and Ser, with a strong preference for Thr on mucin-2, the main constituent of the colon mucus layer (Johansson et al, PNAS, 2010)

(Uniprot: 12 out of 15 O-glycosylations of Muc2 on Thr). Therefore, Thr appears to be the preferred connection residue for O-glycosylations in the colon.

(3) The other amino acids around the Thr/Ser O-glycosylation (assessment done for pos. -10 to +10, Christlet & Veluraja, Biophys J, 2001) did not reveal unique conservations, only preferences for Pro and small side chains. Therefore, including more residues in the simulation would only increase the runtimes, but not reveal meaningful further insights.

L. 319f – Thr aligns with the openings: This is the meaningful orientation in the Gal-GalNAc binding pit of Fap2 such that the polypeptide chain can enter/exit the binding site, which we illustratively wrote in the manuscript using a hook-and-chain analogy.

We added an explanation including the Johansson et al citation (**L. 312f**): **“To gain insight into the binding site of the CRC cell receptor Gal-GalNAc and whether both receptors can bind to Fap2 simultaneously, we docked the Gal-GalNAc disaccharide attached to Thr, as used in our binding experiment (Fig. 3a) and as present in most O-glycosylations in the colon on mucin-2⁴⁹, to Fap2-ECD.”**

P9 line 308-316: First of all, this paragraph should probably be moved to the discussion section. I am also not happy with the explanation of mechanistic detail here – it is all a bit convoluted and does not refer to the figure (that the authors provide later in the discussion).

We agree with the reviewer that parts of this paragraph should appear in the discussion, in particular the second half after a short summary sentence. We however also believe that a short summary of results should end this part of the Results section as done for the other parts, and therefore we left the first sentence there with further elaborations. The second part with our hypothesis that the long rod is a spacer indeed matches best to the discussion. The paragraph here has been re-written, also to conclude our findings specifically for the proposed Fap2/GalGalNAc binding model because we separated the original chapter in two. It now reads as (**L. 365f**): **“In conclusion, the docking and MD simulation of Gal-GalNAc-bound Fap2-ECD together with site-directed mutagenesis revealed not only mechanistic details of the Fn interaction with cancer cells, but also demonstrates the physical separation of two interaction sites on Fap2 for the receptors of different cell types.”**

We added the speculative second part as a sentence to the discussion (**L. 424f**): **“This is achieved through the ~45 nm long rod formed by the Fap2 β -helix that acts as a spacer to keep active NK cells at a safe distance from Fn and invaded tumors.”**

Discussion

P9 line 319-331 – this section would benefit from comparison with other autotransporters from different bacterial species, rather than a collection of toxins

Please see our reply to the “major comment” above – here we deliberately wanted to place the focus on the production of large proteins in *E. coli*.

P10 line 344: why “due to”? Similarly, why in line 346: “in contrast”? The causalities here could be described/explained better. It is not self-evident that multivalent interactions are based on low affinity, for example.

When we wrote the manuscript, the causalities appeared clear to us. Now, thanks to the outside view of reviewer 2, we agree that it appears quite convoluted, especially since we move back and forth from the relevance of the two different Fap2 receptor interactions. We therefore rewrote and rearranged L. 391ff. to: **“The pathophysiological purpose of the Fap2/hTIGIT interaction is to deactivate NK cells and effector T-cells by inhibitory signaling through hTIGIT activation after Fap2 binding. Thereby, transient interaction is likely sufficient to initiate TIGIT signaling in immune cells, as also suggested by the fact that the even weaker interaction of Nectin-2 with hTIGIT (K_D of $\sim 6 \mu\text{M}$)⁵⁴ is sufficient for this⁵⁵. On the other hand, the association of Fap2 to Gal-GalNAc allows colonization of tumors^{36,22} and placenta⁵⁶, which typically requires high affinity. However, individual protein-glycan interactions are typically considered to be weak⁵⁷, and data from us (Fig. 3a) and others³⁶ suggest the same for Fap2 and Gal-GalNAc. Therefore, multivalent interactions are likely to be required. The high density of Gal-GalNAc on the surface of tumor cells and the high level of Fap2 expression in Fn³⁹ provide ideal conditions for this to occur. Notable, although our data indicate different binding sites for the two Fap2 receptors, the possibility of interference between their binding cannot be ruled out due to a potential involvement of hTIGIT glycans and/or conformational changes. Thus, a comparably weak Fap2 affinity to hTIGIT might have an evolutionary advantage when the transient interaction, which is sufficient to deactivate the NK cell activity, is rapidly released and the same Fap2 molecule becomes available for interaction with tumor cells.”**

P10 line 350 please add the affinity values (from literature) here in the text, it would be interesting to compare these to the ones measured in this work.

We added the $K(D)$ value for Nectin2 of ca. $6 \mu\text{M}$ from the citation to allow comparison within the manuscript. This is now in the rewritten paragraph (see comment above).

Methods section:

Please add a section/table with all cloning primers etc used for this study (probably in the section after line 388)

We added the table as SI Table 1 and refer to it in L. 469.

P 11 line 382: custom synthesized DNA (not a “gene sequence”) was purchased here.

We corrected as suggested by the reviewer.

P11 line 401 and following: what antibiotics were added to the overnight culture?

We added this information (L 473).

P13 line 451: what plasmids/constructs were used for expression of hTIGIT in E.coli?

We added details to the pET19-based plasmid, expression and purification were carried out as described by Stengel et al. (L 523): **“We also produced non-glycosylated hTIGIT-ECD in E. coli using a construct that comprises TIGIT residues 23-128 with an N-terminal His6-tag in pET19. Expression and purification from inclusion bodies were carried out as described previously⁵⁹ with the difference that the histag was not removed.”**

P17 line 640: “to a McFarland standard 3” sounds weird, please rephrase. You did not use a standard here.

We thank the reviewer for pointing out this point. Indeed, we did not use a McFarland standard here, but measured the OD to confirm bacterial growth in the overnight culture. We have changed the manuscript text accordingly (L 762): **“After overnight culture and confirmation of bacterial growth by OD measurement, ...”**

P18 line 641: it is not entirely obvious why Jurkat cells are added here to grow Fn. I guess this sentence is misplaced and should be part of the section on cell binding assays. Indeed, Jurkat cells have been added to the *F. nucleatum* culture. It has been shown that the presence of T-cells (Jurkat) stimulates Fap2 expression in *F. nucleatum* (Kaplan et al., 2010). We added the citation and the reason to the Methods section (L 764).

Figures:

Figure 1 vs Figure 2: in Figure 2, not all exemplars of adhesin fibers are kinked. From Figure 1, the reader could get the impression that the kink is a “given”, while from figure 2, one could assume that there are multiple possible conformations. This is not really discussed in the manuscript anywhere? See also supplementary figure 2 where again different conformations are visible.

We agree with the reviewer that the variability in the kink has not been addressed sufficiently by us. Notably, the signal subtraction and local refinement of the longer part was carried out to eliminate variability from the kink that decreased the overall resolution; therefore, there is obviously flexibility. Based on the negative stain (SI Fig. 2d) or cryo (Fig. 1f) micrographs, it is however difficult to assess the kink and its variability, since the images show projections. We therefore created an additional supplementary figure panel in SI Fig. 4b in which we show representative 2D class averages and sort particles according to their kink angles. We determined the total numbers of particles in weakly kinked (i.e. extended) and strongly kinked (i.e. angle close to 90°) 2D classes. While the ~38% of particles are in 2D classes that have a strong kink, 52% appear in more extended 2D classes with kink angles above 110°. The remainder represents “top views” in which the kink angle is not determinable. We added the following sentences to the Results section (L 253): **“A comparison of different 2D class averages revealed variation in the kink angle: the majority of particles (51.5%) is found in 2D class averages with weak kinks, i.e., angles above 110°, whereas 38.2% are in strongly kinked classes with angles close to 90°. The remainder of particles in in top**

view 2D class averages with indistinguishable angles (SI Fig. 4b).”

Reviewer #3 (Remarks to the Author)

The work brings interesting integration between experimental techniques, such as cryogenic electron microscopy (cryo-EM), and computational ones, such as docking and molecular dynamics (MD). The structural analysis of the extracellular domain of the Fap2 adhesin involved cryo-EM, single particle analysis, structural prediction by AI using AlphaFold2, docking and MD, obtaining a 46.5 nm long β -helix. Parameters used for the docking simulations are shown in Supplementary Table 3 and docking and molecular dynamics protocols are detailed in the methods section. However there is little information about the protocol used with AlphaFold2, the authors could extend this topic in the methods section, mentioning more details than the prediction of large proteins piecewise.

We agree with reviewer 3 that more details are required and we now provide information on the design of the fragments of the proteins used for prediction, and the time of the data base used for the prediction, in the methods section (L. 6431). Moreover, we refer to this approach with reference to Methods in the Results section (L. 98).

Another question regarding protocols refers to that used in MD simulations. Several 50 ns replicates were performed for the Fap2-ECD/hTIGIT-ECD and Fap2-ECD/Gal-GalNAc-Thr complexes. The authors could justify the strategy used, considering which would be the best of two scenarios: i) if for the initial structures in each replica different poses obtained from the docking calculations are used or ii) if the same pose obtained from the docking results is used with different initial atomic velocities. Which could be more statistically relevant? Regarding the lengths of the trajectories, would 50 nsec be enough to scan the conformational space to observe the stability of the complexes and be confident about the molecular interactions observed?

We used a combination of both strategies: We selected the most promising docking poses and subjected each to molecular dynamics (MD) simulations with 10 replicates. The MD simulations were performed not only to sample conformational space but also to validate the reliability of the initial docking poses by assessing their stability over time under physiological conditions. For these replicates, we performed separate energy minimizations, which introduced slight variations in the initial conformations of the systems. These variations made the generation of different initial atomic velocities unnecessary, since the initial conformational differences already ensured diverse starting points for the simulations. We added the information to the Methods section under MD simulation.

Details of the initial setup are provided in the manuscript. The Gromacs setup files for energy minimization, temperature and pressure stabilization (em.mdp, nvt.mdp and npt.mdp, respectively) and production simulation (md.mdp) are provided as supplementary data files.

Moreover, we now increased the simulation times to 150 nsec to allow for longer conformational sampling. The corresponding results (L. 2780), figure panels (4d, 5b) and SI movies 2,3,4 were updated with models from the new simulation data, and RMSD plots were provided as SI Fig. 11.

Reviewer #4 (Remarks to the Author)

In this manuscript, Schoepf and colleagues provide structural, and functional analysis of the outer membrane adhesin Fap2 present on the surface of the oral opportunistic pathogen *Fusobacterium nucleatum*, using a heterologous system in *Escherichia coli*. Through cryo-electron microscopy, structure prediction, and modeling, the authors demonstrated that the rod-shaped extracellular region of Fap2 binds to TIGIT, a T-cell protein, and identified a binding pit on the tip of Fap2 for Gal-GalNAc present on cancer cells. Overall, the findings are significant, greatly contributing to understanding of the molecular interactions between *F. nucleatum* and host cells, and the manuscript is well written. However, there are a couple of major issues that need to be addressed for further consideration.

Major comments:

1) While *E. coli* provides a versatile experimental system for this work, it cannot simply replace the native organism *F. nucleatum*, given that genetic tools have been developed for this microbe. There might be potential post-translational modifications in *F. nucleatum* that might affect binding and not be found in this heterologous system. To address this important issue, the authors should perform comparative mass spectrometry analysis (MS/MS) of Fap2 proteins isolated from *E. coli* and native *F. nucleatum* since both are available.

We indeed have both purified Fap2 from *F. nucleatum* ATCC25586 and *E. coli*-produced Fap2 (sequence originating from ATCC23726) available from comparison by mass spectrometry. We carried out a comparison and found no indications for post-translational modifications in the native Fap2 sample from *F. nucleatum* ATCC25586. The data were added to **SI Figure 5** (panel f) and **SI Table 2**. We also refer to the comparison in the Results section (**L 1843**): **“To identify possible differences in post-translational modification (PTM) of recombinantly produced Fap2-ECD with Fap2 in *F. nucleatum*, we digested the proteins and compared the masses of the resulting peptides. Mass spectrometry data revealed that the vast majority of peptides is either unmodified or deamidated, whereas further modifications are of minor occurrence and similar in both Fap2 preparations (SI Fig. 5f). The unannotated mass shift of 60.98 Da in native Fap2 is compatible with bound Na⁺ and K⁺, but no other known PTM (ABRF Delta Mass data base). This indicates that recombinant and native Fap2 have no tractable differences in PTM that could influence their receptor binding activities.”**

Having added the data, we would like to point out that such a comparison has not been done as a routine experiment for recombinantly produced proteins from different organisms, which have been used in structural and biochemical experiments by us and others for many years. We have not found any indications in the literature for specific PTMs in *F. nucleatum*. I would like to use this possibility to ask reviewer 4 if they know of specific PTMs in *Fusobacteria* that we might have missed.

2) The authors should confirm the binding site of Fap2 for Gal-GalNAc by mutational analysis since the binding pit has been identified.

We agree and carried out site-directed mutagenesis. Please find our detailed answer at the second “major comment” of reviewer 1, who also suggested these experiments.

Minor comments:

1) Introduction or relevant sections: more background information on the AIDA autotransporter systems would be helpful.

Following suggestions from reviewer 2, we added more information on the AIDA-I based expression system, including several citations (L 105ff). We also reformulated and extended a sentence about the AIDA-I autotransporter in *E. coli* (L 92ff): **As a prerequisite for these studies, we applied an *E. coli*-derived autotransporter system based on the AIDA-I adhesin, which facilitates diffuse adherence of enteropathogenic *E. coli*^{40,41}. We used pAIDA1^{42,43} as a template to produce Fap2 recombinantly on the OM surface of *E. coli*.”.**

2) Discussion: The authors consider elaborating more on the biological significance of the findings. How are the observed molecular interactions between Fap2 and receptor molecules related to the broader context of fusobacterial roles in cancer and other physiological processes?

In the revised paragraph after L 391 in the discussion, we bring the measured affinity and the two different binding sites into the pathophysiological context, and the last paragraph of the discussion highlights a potential significance of the shape of Fap2 and the positions of the two receptor binding sites.

3) Page 3, line 83: change “46,5” to 46.5” and check the manuscript throughout.

We corrected the typos.

4) Page 3, line 85: please spell out “MD” as first appears.

We spelled out Molecular Dynamics at the place of its first appearance (after removal of the introduction paragraph as requested by reviewer 1 (L 137)).

5) Page 5, line 149: “The triangular extensions at the matchstick region result in two pits at the tip (Fig. 1J), which are only present in Fap2, but not RadD (SI Fig. 1A)”. It is difficult for this reviewer to make this distinction from the secondary structure of RadD alone in SI Fig. 1a alone. Can the authors indicate what two pits are referred to in Fig. 1j?

We agree with reviewer 4 and added an additional SI Fig. 1g, that shows the tips of Fap2 and RadD side by side and indicates the pit in Fap2. A side-by-side representation has been already shown in former SI Fig. 9d, but this is quite restricted and placed too late in the manuscript, therefore we removed SI Fig. 9d in favor of SI Fig. 1g.

6) Page 9, line 314: consider revising to “...keep active NK cells a safe distance from Fn itself”.

Following a suggestion by reviewer 2, we rewrote the last paragraph of the Results section, and we changed the sentence (L 304f) to: **“In summary, our structural assessment of the Fap2-ECD/hTIGIT-ECD complex reveals binding of hTIGIT to Fap2 at the maximally distant position to the Fn surface.”.**

7) It may be helpful to label N/N-termini in Fap2 ECD (Fig. 1g).

As Fig. 1g shows a cryo-EM density map without a fitted model, we labeled the N- and C-termini in the map/model overlay in Fig. 1h.

8) Italicize *E. coli* in SI Fig. 2b and check the manuscript throughout.

We carried out the requested modification.

Reviewer #5 (Remarks to the Author):

Critique 1:

This manuscript presents the cryoEM structures of the adhesin Fap2 and its complex with the T-cell receptor. Fap2 is a pivotal pathogenic factor of Gram-negative *Fusobacterium nucleatum* (Fn). Fap2 extracellular domain (Fap2-ECD), a 400 kDa autotransporter protein, plays a crucial role in binding to cancer and immune cells via two receptors: the glycan Gal-GalNAc and the T-cell protein TIGIT. Given its significant presence in the intestinal microbiome, associated with colon cancer, the third most prevalent cancer globally, Fap2 emerges as an important drug target.

The cryoEM map Fap2 shown in this manuscript is seen as a relatively unique “matchstick” structure—a 50 nm long, rod-shaped molecule with a receptor binding tip. The cryoEM map, resolved at 4.7 Å, fell short of resolving the beta strands, precluding de novo model building. Utilizing AlphaFold2, they proposed a model of a beta-helix with a 1010 kink to fit the density map. Additionally, they resolved the cryoEM structure of the Fap2-ECD complexed with human glycosylated TIGIT-ECD at 6 Å, highlighting the receptor binding site on the distal tip of Fap2ECD through a difference map. Due to the moderate resolution limiting the resolvability of side-chain interactions, they used molecular dynamics to aid in deciphering interactions between Fap2 and TIGIT.

The strengths of this manuscript are the successful expression of the largest membrane-associated protein in *E. coli* and the attainment of subnanometer resolution cryoEM maps. In addition, they used cryoET of cells overexpressing Fap2 to show the matchstick shape of Fap2 to confirm the authenticity of their structural observation of biochemically purified protein. AlphaFold2 and MD simulation were used to build models in elucidating the structure and functional mechanisms governing Fap2’s interactions with tumor and immune cells. However, several weaknesses appear in the manuscript:

1. Admittedly, reporting cryoEM structures in this moderate resolution zone poses challenges for constructing accurate models. Lacking rigorous validation such as a PDB validation report to assess reliability and accuracy places uncertainty on the reporting models.

We thank this reviewer for understanding the challenges of modelling in our maps with limited resolution, especially under the view point that the strands of the β -helix do not appear separated. The map resolution however is true and not an over-estimation of CryoSPARC non-uniform refinement, because (1) a refinement in Relion results in the same resolution (both 4.4 Å, see Methods section, Fig. 200), and (2) a map created from the model at 4.4 Å resolution also

shows no strand separation. We decided to build and share (PDB) the atomic model of Fap2 at the membrane-distal tip region, where we can reliably place and optimize the AlphaFold2 model. We now describe the modeling procedure in the methods (“**Modelling of membrane distal part of Fap2-ECD**”), report the parameters obtained by MolProbity in PHENIX in Table 1, and the map-to-model FSC in SI Figure 3. Although the map-to-model FSC does not reach the map resolution and that because of the limited resolution the model still has some geometry issues (in particular Ramachandran non-favored region and clashes), we decided to share the model together with the experimental map data because we believe that further work in the community can be nucleated by its availability.

2. Detailed residue interactions between Fap2 and the two receptors in Figure 4 rely on docking and MD simulations. I am not convinced that this method of analysis alone can be trusted. Incorporating additional biochemical experimental data or improving the map resolutions would be necessary to support the hypothetical models.

As also suggested by reviewers 1 and 4, we now include binding experiments with Fap2 mutants of the two proposed binding sites. Please find our detailed answer at the second “major comment” of reviewer 1.

In view of the uncertain nature of the models and chemical interactions, the proposed chemical mechanism seems to be somewhat speculative.

Site-directed mutagenesis data (see above) strengthened the binding mechanism for Gal-GalNAc, as originally proposed by docking and MD simulation. The interactions as shown in Fig. 5 are plausible. We however agree that without highly resolved co-structures the mechanisms cannot be stated with certainty, and we therefore added a concluding statement to the corresponding paragraph of the Results section that underlines the importance of further high-resolution structural analysis (L 3471): “**We therefore conclude that these residues and their proposed H-bonds are important for binding, albeit other residues might provide redundancy. Further high-resolution structural research is required to confirm the exact molecular nature of interaction.**”

For TIGIT binding, site-directed mutagenesis did not confirm the binding model proposed through the integrative approach (cryo-EM, docking and MD simulation). Therefore, we describe the limitations and the possibility for alternative binding modes at the same binding site at the end of this chapter in the Results section (L 3053): “**Whereas docking and MD simulation suggest a particular mode of interaction via a TIGIT loop that intercalates into a cleft at Fap2, site directed mutagenesis did not confirm the binding model. Therefore, further structural analysis of the complex at near-atomic resolution is required to understand the binding mechanism of Fap2 to TIGIT at its full extent.**”

Moreover, we added a limitation statement to the discussion (L 4151): “**However, as site-directed mutagenesis of the proposed binding site of Fap2 did not exhibit a significant decrease in affinity, the hydrogen bonds formed by the involved residues are possibly redundantly replaced by others. At this point, our integrated approach reaches its limit and high-resolution data of the complex are required to ultimately visualize the side-chain interactions involved at the binding site.**”

Critique 2:

The authors investigated Fap2, a *Fusobacterium nucleatum* (Fn) adhesin, and its interaction with the glycan Gal-GalNAc and a T-cell protein TIGIT. This is important for understanding the role of Fn in the progression of colorectal cancer.

The authors first recombinantly express Fap2 on the *E. coli* surface and purify Fap2-ECD. A 4-5 Å resolution map was obtained using single-particle cryo-EM analysis. An alphafold2 predicted model was used to fit into the cryo-EM density. A Cryo-ET of native Fn revealed Fap2's native conformation and confirmed that the purified Fap2 has a similar conformation as that of the native Fap2. A fluorescent Gal-GalNAc was used to show binding to Fap2-expressing *E. coli* as well as fluorescent micrographs showing Fap2-expressing *E. coli* binds to cancer cells. SPR was used to confirm purified TIGIT binds to purified Fap2. Finally, the authors also obtained a 6 Å cryo-EM map of Fap2-TIGIT and docked it with the known TIGIT structure. Gal-GalNAc was also docked to the Fap2 model to predict its binding site. MD simulations of both Gal-GalNAc and TIGIT interactions further support the stable bindings.

Comments

- Major

1. The cryo-EM maps are in the 4-6 Å resolution range. Although the model building would be challenging at this resolution, the alphafold2 predicted structure fits relatively well visually. However, it is difficult to judge the quality of the model fitting only by looking at the figures. A metric such as Q-score or other correlation-based methods might be useful to assess the quality of the fitting.

Please see our reply to the first point of critique 1 above. As expected for our low resolution (map – 4.4 Å, model – 6.5 Å), the Q-score is low (average of 0.191), but in most instances above zero (see accompanying PDB validation report). The atom inclusion on the other hand is high (75.4% at medium contour level).

2. It would be nice to demonstrate the quality of the map and model fitting by zooming into some local area such as beta sheets or the N terminal tip.

We added parts of the model fitted to the map of the locally refined Fap2 matchstick to SI Fig. 3.

3. Supp fig. 4: Why does it require a density map generated from the alphafold2 model as an initial model for non-uniform refinement? A 5 Å filter is not sufficient to prevent model bias. In addition, this step is probably not necessary as the map (4.79 Å, 265,956 ptcls) is visually similar and with a similar resolution as the map without the generated initial map (4.73 Å, 265,956 ptcls).

The unique shape and the kink of Fap2 raised an issue in data processing: upon 3D hetero refinement, we ended up with “shorter” maps, in which “tilted side views” were interpreted solely as side views from the program (see the image below). Therefore, we added this control to our processing strategy and compared the resulting length of the Fap2 rod in the cryo-EM map with the length of the cryo-EM map before the aforementioned 3D hetero refinement that did not have this bias. However, this additional step did neither improve the resolution nor reveal an additional, longer species, showing that the originally reconstructed and refined map was already correct. We however left this processing step in the description, since the map of the tip of the longer part was based on refinements after this step, although the outcome was the same.

Comparison of map lengths of Fap2 with different processing steps. The red dashed lines indicate the upper and lower borders of the longest map (gray). The left, longest map is the one where the Fap2 model was fitted. The other, shorter maps were the result of different processing steps, in particular the selection of 2D classes without the longest classes (purple, green), or after a 3D hetero refinement (blue). Note that all four maps are shown in the same view with their maximum length.

We agree with reviewer 5 that the reason for this unconventional processing step should be explained, and added the following information to **SI Fig. 4** legend: “**Note an unconventional data processing step that included the verification of the resulting Fap2-ECD map (full-length, left side) with a map generated from an AlphaFold2 model. This is required because of the unique shape of Fap2 – slightly tilted orientations of side views could otherwise be interpreted as full side views and result in shortened reconstructions. This was occurring in Heterogeneous refinements in CryoSPARC.**”

- Minor

1. Line 131 and 135: Not clear what “4,5 nm” and “1,2 nm” mean.

We corrected the typos to 4.5 nm and 1.2 nm.

2. Line 138: Indicate where the “two triangular extensions” are in Fig. 1g.

We indicated the extensions as suggested by highlighting them with red dashed lines.

3. Line 139: Indicate where are the residues K525-K836 in Fig. 1h.

We now depict these residues that form the extensions (see above) in a different color (blue) in the model and describe this in the figure legend.

4. “Longer branch”, “tip of the longer branch”, and “matchstick region” were used randomly in the manuscript. It’s not clear what they mean. Please clarify and use them consistently.

We thank reviewer 5 for identifying this inconsistency. Reviewer 2 already pointed out that “branch” is the incorrect notion here. We therefore do not use the word “branch” anymore in the manuscript and changed “tip of the longer branch” to “matchstick region” consistently.

5. Fig. 1d should color the amphipathic helix and unstructured N-terminal region similar to that in Fig. 1b to make it clear that the two regions are in the construct.

We changed the color coding in Fig. 1d, also in response to a comment of reviewer 1.

Reviewer #6 (Remarks to the Author):

Thank you for participating in this important initiative.

We thank Michael Kolbe and the other anonymous reviewers for their helpful comments that guided us to improve the manuscript. Please find our replies to the last remaining comments below in blue. Changes and additions in the main manuscript text are highlighted yellow.

Reviewer #1 (Remarks to the Author):

Thanks a lot for providing a detailed reply addressing all the points raised during the review. I suggest that you either replace the SDS-PAGE shown in Fig1 with the gradient gel from the reply or you add the gradient gel as supplementary data. Very nice work!

Michael Kolbe

We thank M. Kolbe for the appreciation of our work. A section of the gradient gel that we presented in the last revision document has been added as **Supplementary Fig. 2c**, and the full figure is shown in the Source data file.

Reviewer #2 (Remarks to the Author):

all of my comments have been addressed. The reviewed version reads well.

We thank reviewer 2 for the acknowledgement of our work.

Reviewer #5 (Remarks to the Author):

The authors have made significant improvements in the revised manuscript, enhancing readability and presenting their findings more clearly. In this version, they provide additional evidence supporting their original hypothesis that *Fusobacterium nucleatum* adhesin Fap2 binds to Gal-GalNAc on colorectal cancer (CRC) cells and T-cell protein TIGIT, contributing to colorectal cancer progression.

Notably, the authors generated an additional mutant to abolish Gal-GalNAc binding, guided by their docking model. The results convincingly demonstrate that the predicted interactions are critical for binding. However, the mutant designed to disrupt hTIGIT binding did not validate the predicted interactions, leaving the precise nature of the Fap2-hTIGIT interaction unresolved.

The authors have addressed all my comments from the first round of review. However, I have a few additional comments based on the revised manuscript:

Comments:

Could the authors clarify why a negative control with *E. coli* (similar to Figure 3b) was not included in Figure 4e? This would help confirm if both WT and mutant bind hTIGIT.

The goal of the cellular interaction assay in Fig.4e was to directly quantify the difference in binding between WT and mutant Fap2 to hTIGIT-expressing HEK293 cells. As cell binding of *E. coli* that

express mutant Fap2 was slightly higher than of those that express WT Fap2, we were able to conclude that this particular mutant does not result in impaired binding without using a control of Fap2 non-expressing *E. coli*. We confirmed in independent pulldown assays using purified proteins that both WT and mutant Fap2 bind to hTIGIT, shown in Supplementary Fig. 9a.

Given the negative results in Figure 4e and Supplementary Figure 9a, along with the low resolution (~12 Å) of the cryo-EM map at the hTIGIT interaction region, the docking-based interaction model in Figure 4d should be interpreted with caution. The authors may consider moderating their description (Lines 271-287) to reflect these uncertainties.

We agree with reviewer 5 and re-formulated a central sentence in the description of the preceding docking and MD simulation experiments in l. 271ff to remove the words “validate”, “confirmed” and “stable” that would suggest an experimentally confirmed interaction site:

“Due to the limited resolution of the cryo-EM map, we applied an integrative approach to model the hTIGIT-ECD (residues 23 - 128) on Fap2. For this, we first identified residues of Fap2 that are in close spatial proximity to the additional density in the complex and then docked the polypeptide part of hTIGIT-ECD (PDB 3UCR) to this Fap2 part using HADDOCK⁵⁸ (Supplementary Table 4). The best cluster of docking poses (score: -14.6 +/- 6.7) revealed that loop P87-G92 on hTIGIT-ECD intercalates into a cleft at the Fap2 tip that is formed between D360, T397, R398, E425, N494, D496, and W540 (Fig. 4c,d). We then tested the docked models in MD simulations. In 7 out of 10 replicates, hTIGIT remained bound throughout the 150-ns trajectories (Supplementary Movie 3). Fap2-W540 forms a hydrogen bond with hTIGIT-P65 and simultaneously interacts via hydrophobic contacts with P67, which intercalates deeply into the cleft at the Fap2 tip. Hydrogen bonds between Fap2-W540, R427, S371, K368 and R398, and hTIGIT-Q22, H24 A30, V32 and P65 are identified as the major contributors to the interaction (Fig. 4d, Supplementary Table 5). Fap2-R427 forms a hydrogen bond with the hTIGIT-Q22 and H24 side chains. Fap2-N541, S371 and K368 form a C-shaped interaction interface and participate in consistent hydrogen bonds with hTIGIT-W53, A30 and V32, respectively, throughout the MD simulation replicates in which the complex remained intact”

Moreover, the statement at the end of the chapter (l.304ff) already addresses the fact that site-directed mutagenesis did not agree with the docking and MD simulation results. We modified the last sentence to advise interpreting the docking and MD simulation results with caution: **“Therefore, the docking and MD simulation results should be interpreted with caution and further structural analysis of the complex at near-atomic resolution is required to understand the binding mechanism of Fap2 to hTIGIT at its full extent.”**

Could the authors provide a cryo-EM map movie for the Fap2/TIGIT complex, similar to the one provided for Fap2 alone? This would allow readers to better assess the map quality and set appropriate expectations for the accuracy of the predicted interactions in Figure 4d.

We thank the reviewer for the suggestion and now provide this movie as Supplementary Movie 2 (referred to in the Results part under “hTIGIT binds to the membrane-distal tip of Fap2”).